# Identification and characterization of foehn events in Beijing and their impact on air-pollution episodes

Ju Li[1,2,3], Jingjiang Zhang[1,2], Mengxin Bai[4], Jie Su[1,2], Qingchun Li[1,2], Xingcan Jia[1,2]

[1] Institute of Urban Meteorology, CMA, Beijing, 100089, China
[2] Beijing Research Center for Urban Meteorological Engineering and Technology, Beijing, 100089, China
[3] State Key Laboratory of Severe Weather, Chinese Academy of Meteorological Sciences, Beijing, 100081, China
[4] Beijing Municipal Climate Center, Beijing, 100089, China

*Correspondence to*: Ju Li (jli@ium.cn)

**Abstract.** This study proposes a method for identifying foehn events in Beijing using automatic weather station (AWS) data, considering upper-air wind direction, topography, meteorological changes, and foehn propagation. Analysis of AWS data from 2015 to 2020 revealed an annual average of 56.5 foehn days, with these days occurring most frequently in winter and least frequently in summer. High-frequency foehn areas exhibit a band-like distribution from the northwestern mountainous region to the southeastern plains, while low-frequency areas are primarily concentrated in the northeastern plains. The horizontal extent of the foehn influence is maximal in spring and minimal in summer. Foehn-induced hourly temperature increases can exceed 11 °C, peaking from night to early morning. Approximately 67% of pollution episodes are accompanied by foehn events, with foehn duration negatively correlated to pollution episode duration. 60.4% of foehn events coincide with decreasing concentrations of particulate matter of 2.5 μm diameter (PM2.5), while 39.6% show increases. Rapid PM2.5 concentration increases (> 50 μg m$^{-3}$/h) primarily correspond to weak foehn events (temperature increase < 2 °C). Foehn winds influence pollution through direct and indirect effects. The direct effect, associated with strong northwesterly pressure gradients, can rapidly decrease pollutant concentrations. The indirect effect, linked to weak pressure gradients, alters the boundary-layer structure, causing rapid increases in pollutant concentrations following the termination of foehn. This foehn identification method, applicable to long-term historical surface observations, not only facilitates a deeper understanding of how foehn phenomena evolve and contribute to temperature increases under global warming, but also advances an in-depth exploration of the relationships between foehn events and high-impact weather phenomena.

## 1 Introduction

Foehn winds are local dry, warm winds occurring on the leeward side of mountains, resulting from descending air flows. They are characterized by warm and dry air, often accompanied by strong gusts and a significant reduction in cloud cover on the leeward side of mountain ranges (Brinkmann, 1971; Richner and Hächler, 2013). The term "foehn" originally referred to a warm, dry wind formed in German, Austrian, and Swiss valleys after air flows crossed the Alps (Whiteman, 2000). Other regional foehn-type winds include the Chinook winds on the eastern side of the Rocky Mountains in the United States (Brinkmann, 1974; Durran, 1986) and the Santa Ana winds in Southern California (Raphael, 2003; Guzman-Morales et al., 2016; Rolinski et al., 2019). Foehn winds occur on the leeward slopes of most major mountain ranges worldwide and have been extensively studied. Examples include foehn winds in the Alps (Hoinka, 1985a,b; Gohm and Mayr, 2004; Jaubert et al., 2005; Drobinski et al., 2007; Cetti and Sprenger, 2015; Haid et al., 2020), Japan (Kusaka and Fudeyasu, 2017), New Zealand (McGowan and Sturman, 1996; McGowan et al., 2002), and the Antarctic Peninsula (Orr et al., 2008; Elvidge et al., 2016; Turton et al., 2018; Elvidge et al., 2020). These dry, warm winds impact agriculture, ecosystems, and climate systems, affecting plant growth and development (Walker and Ruffner, 1998) and increasing the risk of avalanches, floods, and glacier melting (Barry 2008; Cook et al. 2005; Kuipers Munneke et al., 2012). Strong gusts associated with foehn winds can damage buildings and property, potentially triggering and rapidly spreading wildfires (Westerling et al. 2004; Sharples et al. 2010). Foehn winds can also exacerbate the effects of heatwaves (Takane and Kusaka, 2011; Nishi and Kusaka, 2019; Nishi et al., 2019; Lian et al., 2008) and influence air-pollution levels by affecting pollutant transport and altering the boundary-layer structure (Li et al., 2015; Li et al., 2020).

The formation of foehn winds is commonly attributed to terrain-induced latent heat release and precipitation mechanisms, which are widely adopted in textbooks. Currently, four main mechanisms are recognized in the academic community (Seibert et al., 1990; Ólafsson, 2005; Elvidge and Renfrew, 2016): isentropic drawdown, latent heat release and precipitation, mechanical mixing, and radiative heating. Miltenberger et al. (2016) found that thermodynamic effects dominate foehn formation in Switzerland, while Seibert (1990) and Würsch and Sprenger (2015) showed that dynamic effects contribute more significantly. Kusaka et al. (2021) reported that 80.8% of foehn events in Japan occurred without precipitation, suggesting that thermodynamic effects are not always dominant. Foehn formation depends on various factors, including local geography, topography, and weather conditions, and can result from single or multiple mechanisms. Therefore, when studying foehn causes, it is necessary to conduct detailed and comprehensive analyses considering the specific geographical and weather conditions of the research area. Foehn identification methods vary depending on the region and research objectives. A simple approach is to classify days with high temperatures, low humidity, and winds from mountainous areas as foehn days (Shibata et al., 2010); however, this method may misidentify large-scale phenomena as foehn events. Most methods require hourly temperature increases of at least 1 °C, specific surface wind directions, and decreased humidity. Some methods also include wind speed thresholds and quantitative humidity reduction requirements (Speirs, 2012), while others consider both surface and upper-air wind direction and speed (Kusaka et al., 2021). In addition to surface meteorological observations, many studies utilize reanalysis data and radar observations for foehn identification and trajectory tracking (Kusaka et al., 2021; Jansing et al., 2022).

The eastern foothills of the Taihang Mountains are prone to foehn winds, which have extensive impacts on the North China Plain's agricultural production, heatwaves, and air pollution. Consequently,

Taihang Mountain foehn events have attracted significant research attention and are one of the hotspots
in Chinese foehn research (Zhao et al., 1993; Xiong et al., 2020; Wang et al., 2012a, 2012b). Various
identification methods have been developed for Taihang Mountain foehn, such as those proposed by
Zhao et al. (1993), Wang et al. (2012a), and Xiong et al. (2020). However, these studies primarily focus
on the central and southern sections of the Taihang Mountains. Beijing's main urban area and population
are concentrated in the plain formed by the intersection of the northern Taihang Mountains and the
Yanshan Mountains (also known as the "Beijing Bay"), which is susceptible to Taihang Mountain foehn
winds. Due to the distinct environmental differences between Beijing and the central and southern
Taihang Mountains, existing foehn identification methods and derived climatic characteristics may not
accurately represent the foehn winds affecting Beijing. Therefore, it is necessary to develop a foehn
identification method specifically tailored to Beijing's unique geographical environment and weather
conditions, and to conduct long-term foehn characteristic analysis based on this method.
Foehn winds can influence the transport and distribution of atmospheric pollutants. For example, the
collision of foehn winds with valley winds in canyon topography can lead to severe air-pollution events
(Li et al., 2015), and foehn winds can cause horizontal and vertical transport of ozone (Seibert et al.,
2000). The North China Plain, east of the Taihang Mountains, is one of China's most severely air-
polluted regions. The area's severe air pollution problems are related to high local pollution emissions
(Zhao et al., 2012) and complex terrain, land use, and land cover that induce local circulations such as
mountain–valley winds, sea–land breezes, and urban heat-island circulations (Liu et al., 2009; Wang et
al., 2017). These factors influence pollutant transport and lead to severe air-pollution events (Zheng et
al., 2015; Sun et al., 2016; Ma et al., 2017). Despite this, there have been few studies on the impact of
foehn events on air pollution in this region. Yang et al. (2008) analyzed the effects of Taihang Mountain
foehn winds on PM2.5 concentrations, finding that foehn winds can reduce PM2.5 concentrations and
increase visibility in plain areas. Li et al. (2020) proposed that foehn winds can indirectly exacerbate air
pollution based on an analysis of a pollution process with a haze front and discovered a close connection
between foehn events and pollution events. However, due to the lack of analysis of more pollution events,
there is insufficient understanding of the relationship between foehn winds and pollution events. It is
necessary to utilize observational data from a wider range and longer time series to study the relationship
between foehn winds and pollution events, further revealing the impact and mechanisms of foehn winds
on air pollution.
The objective of this paper is to establish a foehn identification method for the Beijing area based on
AWS data, conduct foehn characteristic analysis, and investigate the relationship between foehn winds
and pollution events. The article is divided into seven chapters. Following the introduction, Chapter 2
introduces the data and methods used, Chapter 3 focuses on foehn identification, and Chapter 4 presents
statistical analysis of foehn characteristics. The relationship between pollution events and foehn winds
is explored in Chapter 5. Chapter 6 provides a discussion, and conclusions are presented in Chapter 7.
**2 Data and methods**
**2.1 Data**
Meteorological data used in this study comprise hourly observations from all operational Automatic
Weather Stations (AWSs) in the Beijing area from 2015 to 2020. The observed elements include
temperature, relative humidity, pressure, precipitation, and 2-minute average wind direction and speed.

AWSs were categorized into Plain AWSs (PAWSs, elevation ≤ 200 meters) and mountain AWSs (Non-PAWSs, elevation > 200 meters). Among the mountain stations, Foyeding Station (FYD, elevation 1224.9 meters) was selected as the representative High-Mountain AWS (HMAWS). Located on a mountaintop at the northwestern border of Beijing and Hebei Province, its wind measurements are approximately representative of upper-air winds at around 900 hPa. Figure 1 illustrates the distribution of AWSs used in this study. Among the plain stations, 14 are national stations, while the rest are regional stations. National stations are installed in standard meteorological fields, compliant with WMO observation regulations, providing better observational environments and higher data quality, as well as more continuous data compared to regional stations. All national stations have observational data for the selected 6-year period, while some regional stations lack data for earlier years, as they were not yet established. Based on their proximity to mountainous areas, plain national stations were further classified into Near-Mountain Plain National AWSs (NM-PNAWS, large blue dots in Fig. 1, totaling six stations) and Non-Near-Mountain Plain National AWSs (large orange and purple dots in Fig. 1).

Air-pollution data consist of hourly PM2.5 concentration values from 33 environmental monitoring stations (white triangles in Fig. 1) within Beijing, published by the Ministry of Ecology and Environment. The data cover the same time range as the meteorological data and can be downloaded from https://quotsoft.net/air/. The hourly average PM2.5 concentration across the 33 environmental monitoring stations was calculated to obtain a city-wide average PM2.5 concentration time series. Continuous periods with city-wide average PM2.5 concentrations exceeding 35 μg m⁻³ and a mean value greater than 75 μg m⁻³ were defined as pollution episodes. The PM2.5 thresholds employed to define pollution episodes in our study are derived from China's Ambient Air Quality Standards (GB 3095-2012; Ministry of Environmental Protection, 2012a) and the Technical Regulation on Ambient Air Quality Index (on trial) (HJ 633-2012; Ministry of Environmental Protection, 2012b). The selection of 35 μg m⁻³ as the baseline threshold aligns with the regulatory transition from "excellent" (Class I) to "good" (Class II) air quality, while 75 μg m⁻³ reflects the onset of pollution episodes (Class III or higher) under China's air quality classification framework.

Sea-level-pressure (SLP) data from the European Centre for Medium-Range Weather Forecasts (ECMWF) ERA5 reanalysis were used to determine weather patterns associated with different foehn types during pollution episodes. Data with a horizontal resolution of $0.25° \times 0.25°$ covering latitudes 32°N to 51°N and longitudes 100°E to 130°E were utilized, and Self-Organizing Maps (SOMs) were employed for weather pattern classification.

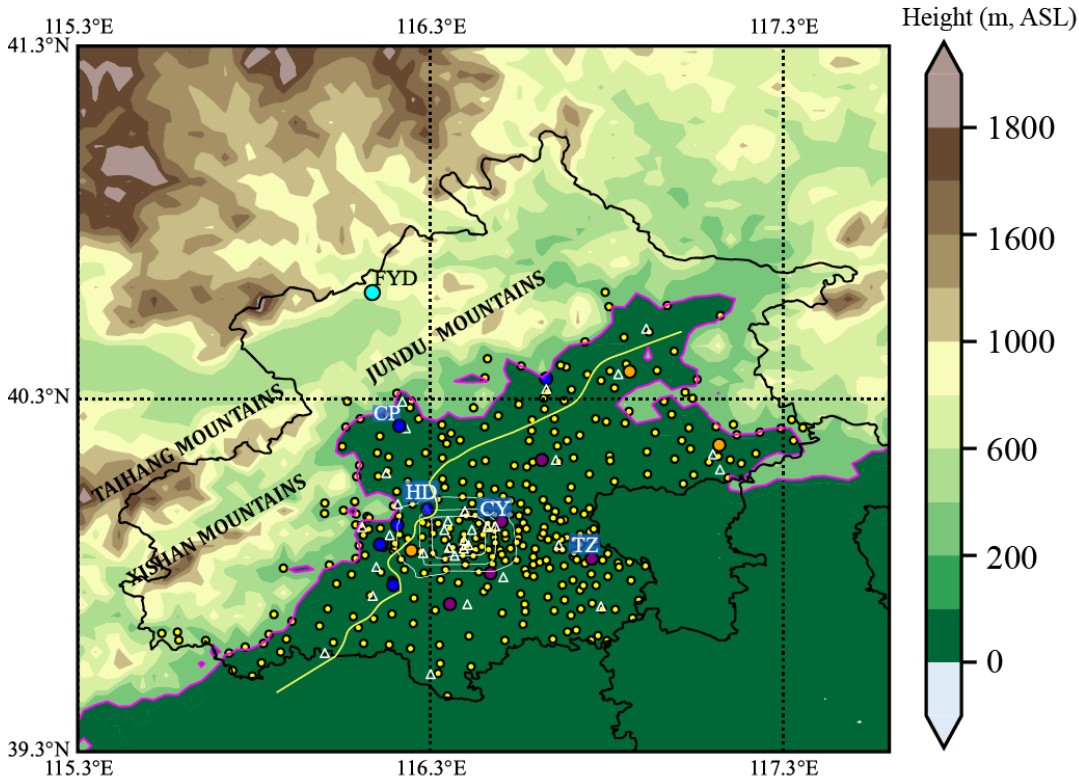

148

**Figure 1: Distribution of observation sites in Beijing, China. The map shows the locations of various Automatic Weather Stations (AWSs): small yellow dots represent Regional AWSs situated at elevations below 200 meters. The large light-blue dot indicates the High-Mountain Station at FYD. Large dark-blue dots represent the Near-Mountain Plain National AWSs (NM-PNAWSs). Large purple dots denote the National AWSs in the central and eastern plain areas. Large orange dots mark other National AWSs in the plain area. Some key National AWSs are labeled with their name abbreviations. White triangles represent air-pollution monitoring stations. The white concentric circles respectively represent the Third, Fourth, and Fifth Ring Roads. The pink lines indicate the contour line at an elevation of 200 m. The AWSs located between the pink and yellow lines are stations selected as the Near-Mountain Plain AWSs (NM-PAWS).**

**2.2 Methods**

For weather pattern classification associated with different foehn types during pollution episodes, we applied the SOM method (Kohonen, 1995). SOM has been widely applied in meteorological research (Rolinski et al., 2019; Ohba and Sugimoto, 2020; Liao et al., 2020) and in classifying weather patterns associated with foehn winds (Kusaka et al., 2021). This method comprises a neural network that uses unsupervised learning to produce low-dimensional representations of high-dimensional input vectors. SOMs consist of input and output (competitive) layers, mapping high-dimensional samples from the input layer to one- or two-dimensional grids in the output layer. The number of output layer nodes equals the number of clusters (N). For different pollution stages (on a daily basis), SLP data from ERA5 were used to train the SOM model. We used 9317 SLP grid points, with the SLP spatial field for each pollution day serving as a vector field. The input layer was set to *m* samples (80 and 33 for the Type I and Type II

foehn winds during pollution episodes, respectively). The input pattern can be denoted as X = {$x_i$: $i$ =
1, ..., $m$}; the output layer contains $n$ neurons, denoted as Y= {$y_j$: $j$ = 1, ..., $n$}; and the connection weight
between input unit $i$ and output layer neuron $j$ in the computational layer can be written as W$_j$ = {$w_{ji}$: $j$ =
1, ..., $n$; $i$ = 1, ..., $m$}. The mapping relationship between the two is given by equation (1):
$$Y = XW, \quad (1)$$

During sample training, only one of the $n$ output neurons is optimal, with its weight given by equation
178  (2):

$$\Delta w_{ji} = \eta \cdot (x_i - w_{ji}) Y_j. \quad (2)$$

where η is the number of training iterations, set to 10,000 in this study. Through weight optimization,
the weight vector of the optimal neuron is moved towards the selected input sample. This training
iteration process is repeated until convergence, ultimately achieving the learning objective. The
determination of the optimal number of nodes is based on two main considerations. First, we focused on
minimizing quantization error while maintaining the interpretability of the identified patterns.
Preliminary tests with node numbers ranging from 3 to 8 showed that when the node count exceeded 6
for Type I or 4 for Type II, the resulting clusters became overly fragmented, lacking meaningful
meteorological distinctions. Second, expert evaluation confirmed that only 6 nodes for Type I and 4
nodes for Type II effectively separated distinct SLP configurations, such as changes in troughs and ridges.
These configurations provided a clear and interpretable distinction between meteorological patterns.
Based on these two factors—quantization error minimization and expert validation—we concluded that
6 nodes for Type I and 4 nodes for Type II are optimal for accurately capturing the relevant
meteorological features.
**3. Identification of foehn events**
Our objective is to develop a method for identifying foehn events based entirely on AWS data. The
advantage of this method is that it allows for the identification of foehn events using the same type of
observational data over longer time series, facilitating long-term climatic analysis and research of foehn
winds. According to the characteristics of foehn winds in the Taihang Mountains (Wang et al., 2012a),
the formation of foehn winds in this region requires a background wind from the northwest at high
altitudes, with the wind direction roughly perpendicular to the southwest–northeast orientation of the
Taihang Mountains. Additionally, the occurrence of a foehn event follows a specific temporal sequence:
it first appears in the plain areas near the leeward slope and then sequentially at downstream locations
along the foehn propagation path. Therefore, the National Meteorological Station FYD, at an elevation
of 1224.9 meters, was selected as the high-altitude wind observation station. This choice avoids issues
such as shorter observation periods and data format inconsistencies that can arise from using other high-
altitude wind observation data, such as wind profiler radar.
By studying 22 representative historical foehn cases, we developed a method for identifying foehn
events in the Beijing area based on AWS data (Fig. 2). First, we determine whether a specific station
within NM-PNAWS is experiencing a foehn event. If the following conditions are met simultaneously
at a given time, then this time is considered a foehn hour at this NM-PNAWS: the wind direction at FYD

is northwest (270–360°), there is no precipitation at any plain station, the temperature at this NM-PNAWS exceeds the mean temperature of the representative stations in the Central and Eastern PAWS (CE-PAWSs), the wind direction at NM-PNAWS is 250–360° or 0–45°, the hourly temperature change is greater than 1 °C, and the hourly relative humidity change is negative. Foehn winds are dry, warm downslope winds generated by subsidence on the leeward side of mountains, requiring upstream airflow to intersect with mountain barriers at a sufficient angle to induce lee-side descent rather than parallel flow. The 250–360° range corresponds to airflow traversing the Taihang Mountains, Xishan Mountains, and Jundu Mountains (Fig. 1), ensuring an intersection angle between wind direction and mountain orientation. The 0–45° range targets northeasterly winds interacting specifically with the Jundu Mountains. By adopting a 1-hour warming threshold >1℃ (consistent with prior criteria), NM-PNAWSs can consistently capture the top 15% (≥85th percentile) of high-warming data over 24 hours. Increasing the threshold above 1℃ reduces data inclusion—notably at night—whereas decreasing it below 1℃ introduces excessive weak warming events. The condition that the temperature at NM-PNAWS must be higher than the average temperature at CE-PAWS is introduced to select the moments at which the temperature rise at this station precedes that at CE-PAWS. If at least two foehn hours occur at the same NM-PNAWS on the same day, that day is defined as a foehn day for that station. If at least two NM-PNAWSs experience foehn days on the same date, that date is defined as a city-wide foehn day. Identification of a single-station foehn for other plain stations that are not NM-PNAWS is only conducted on city-wide foehn days. This involves sequentially identifying single-station foehn hours and station foehn days, as detailed in Fig. 2.

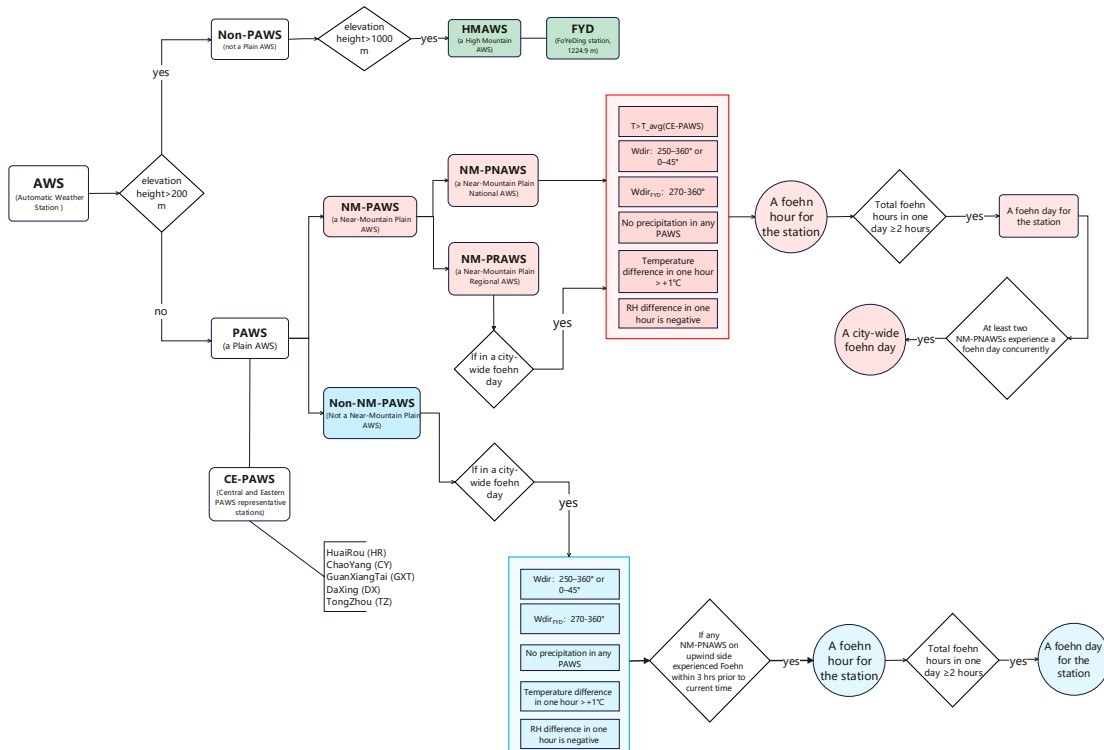

**Figure 2: Flowchart of foehn identification based on AWS data.**

## 4. Analysis of foehn characteristics

Based on the aforementioned methodology, foehn days at all PAWSs in Beijing were identified for the

period from January 1, 2015 to December 31, 2020. The temporal variation of foehn days across all PAWSs in the Beijing region over six years is summarized (Table 1). The six-year average number of foehn days for all PAWSs is 56.5, with notable differences in both the annual mean and maximum foehn days among years, exhibiting an undulating trend over time. The highest average was observed in 2016 with 64.4 days, while the lowest occurred in 2017 with 47.6 days. The maximum number of foehn days peaked at 118 days in 2020 and bottomed out at 90 days in 2015.

**Table 1.** Annual statistics of foehn days at Plain AWSs in the Beijing area.

| Year | 2015 | 2016 | 2017 | 2018 | 2019 | 2020 |
|---|---|---|---|---|---|---|
| Annual average number of foehn days | 51.7 | 64.4 | 47.6 | 52.9 | 62.0 | 59.9 |
| Annual maximum number of foehn days | 90.0 | 105.0 | 108.0 | 110.0 | 115.0 | 118.0 |

Figure 3 illustrates the annual cumulative distribution of foehn days at PAWSs, revealing a generally consistent horizontal distribution pattern across different years. High-frequency foehn zones are roughly aligned in a northwest-to-southeast direction. This band-like distribution of foehn days is associated with the alignment and topographic configuration of the Jundu Mountains, Taihang Mountains, and Xishan Mountains (Fig. 1). Additionally, the frequent occurrence of foehn events may be linked to topographic gaps. Specifically, the junction of the Jundu and Taihang Mountains contains multiple gaps and valleys. Studies in the Alps have shown that many foehn events occur downstream of such gaps, which is attributed to the transition of airflow from subcritical to supercritical flow as it passes through the gaps. This transition generates strong subsidence and turbulent mixing: during subsidence, air warms due to adiabatic compression and experiences reduced relative humidity, ultimately leading to foehn conditions (Mayr et al., 2007). Mountain-proximal regions, specifically the western and northwestern parts of Changping District, the western portion of Haidian District, the western section of Mentougou District, Shijingshan District, and parts of the western area of Fangshan District, are characterized as high-frequency foehn occurrence zones. Conversely, areas with fewer foehn days are predominantly found in the northeastern plain of Beijing (in the vicinity of Miyun District). Additionally, some urban areas within the Fifth Ring Road also experience relatively low frequencies of foehn days.

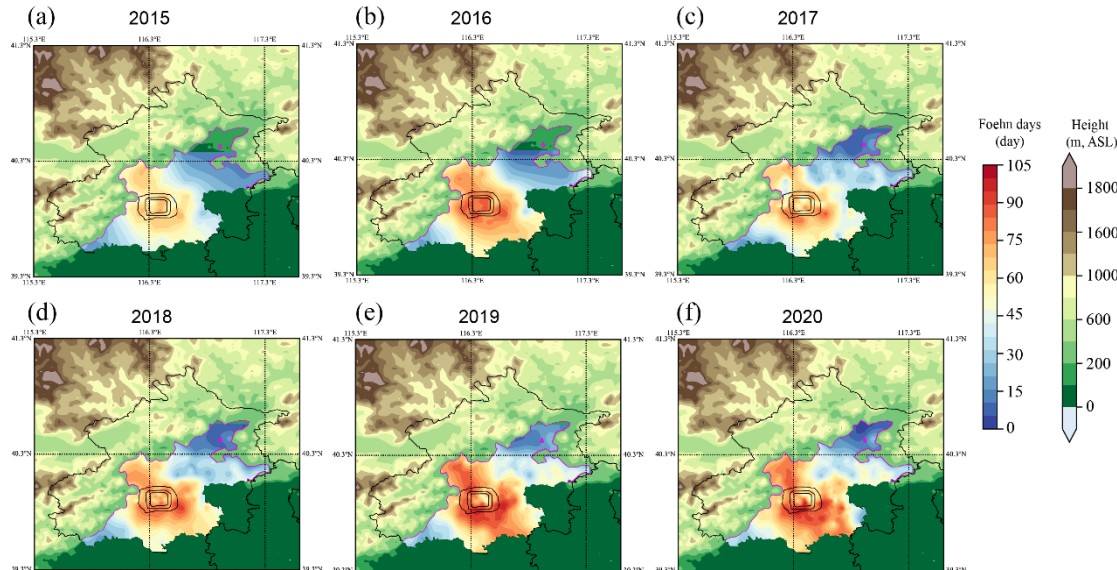

**Figure 3: Annual distribution of foehn days. The pink lines indicate the contour line at an elevation of 200 m.**

To represent the spatial extent of foehn effects, Figure 4 presents violin plots depicting the proportion of stations experiencing foehn winds within the plain areas for each year. More than 50% of the foehn days saw the impact extend over 60% of the stations. There exists inter-annual variability in the horizontal reach of foehn winds. Most years exhibit a unimodal "spindle-shaped" violin plot with a prominent midsection, suggesting a more concentrated distribution of foehn influence within specific ranges. Notably, 2015 and 2016 demonstrated more extensive foehn impacts, with their peak station percentage exceedance surpassing 70%. In 2020, while the distribution maintained a spindle shape, it lacked a distinct peak; the majority of samples fell within the 50% to 70% interval without a clear modal value, marking the lowest median across the six years. Analyzing the yearly medians suggests an overall trend of a narrowing foehn impact scope over time.

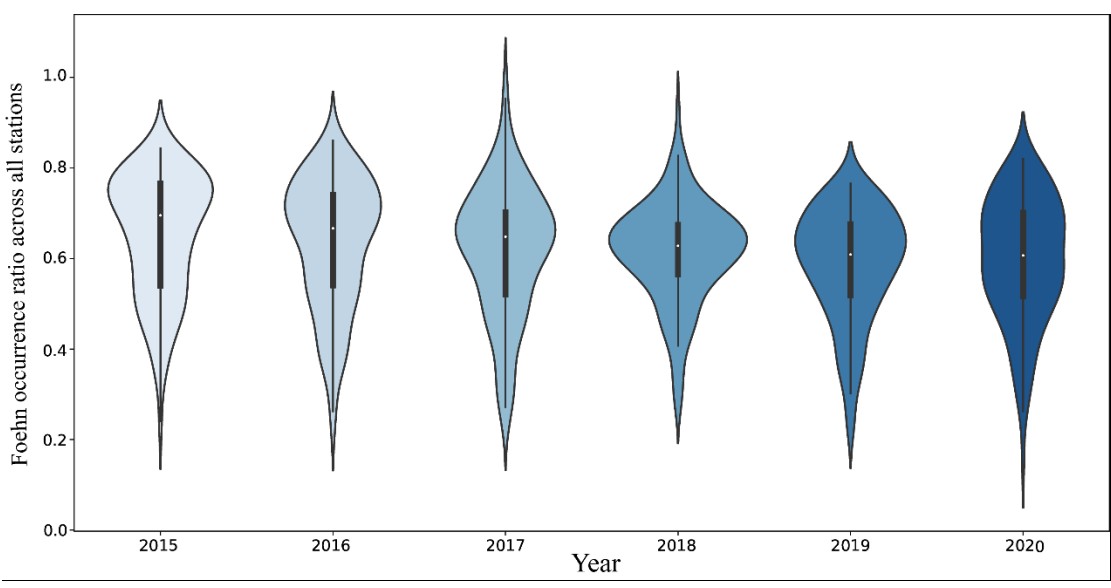

**Figure 4: Violin plots depicting the proportion of PAWSs experiencing foehn winds for each year.**


Table 2 compiles the monthly counts of foehn days for all PAWSs in the Beijing region. The multi-
year monthly average peaks in January with 8.6 days and bottoms out in July with 1.1 days. The monthly
maximum number of foehn days reaches its apex in January with 16 days and reaches its nadir in July
with 2.5 days. Seasonally, winter sees the highest frequency of foehn days, followed by spring and
autumn, with summer having the least.
**Table 2.** Monthly statistics of foehn days at PAWSs in the Beijing area.

| Month | 1 | 2 | 3 | 4 | 5 | 6 | 7 | 8 | 9 | 10 | 11 | 12 |
|---|---|---|---|---|---|---|---|---|---|---|---|---|
| Monthly average number of foehn days | 8.6 | 7.6 | 6.7 | 5 | 5.6 | 3.2 | 1.1 | 3.3 | 5.1 | 5 | 4.3 | 6.2 |
| Monthly maximum number of foehn days | 16 | 13 | 12.3 | 8.3 | 9.7 | 5.7 | 2.5 | 5.8 | 9.3 | 8.7 | 8.8 | 11.8 |


Marked disparities are evident in the seasonal variation of the horizontal distribution of foehn days.
While the general pattern of high-value zones for the monthly average foehn days resembles that of the
annual total, individual months exhibit differing ranges, leading to discernible discrepancies in their
horizontal distribution forms (Fig. 5). January, April, July, and October are selected to represent their
four respective seasons. Overall, foehn day frequencies peak in winter, followed by spring and autumn,
with summer witnessing the least. The high frequency of foehn events in winter is related to the cold
high-pressure systems coming from the northwest. More stable atmospheric stratifications, combined
with the intrusion of cold high-pressure systems, are conducive to the formation of lee waves, which in
turn generate foehn winds. These foehn events typically occur during and at the end of pollution episodes
(which will be further analyzed in the subsequent sections).
In terms of horizontal distribution, winter's foehn days feature two high-value zones, one in the central
urban district and another beyond the southeastern Fifth Ring Road, with the latter recording the highest
values. Spring identifies three high-value zones: the mountain-adjacent interface of Changping District
and Haidian District, the southwestern part of the central city, and again beyond the southeastern Fifth
Ring Road, with the maximum located in the southwestern corner of the central city. Autumn also
highlights three high-value zones, mirroring those of spring but with slightly higher values around the
Changping–Haidian mountain interface and southwestern Fifth Ring Road outskirts. Summer discerns
two high-value zones in the northeastern central city and south of the Fifth Ring Road, with the southern
periphery recording the highest. Regarding the monthly variation in the extent of the foehn influence
(Fig. 6), April experiences the broadest impact, with July witnessing the narrowest. The seasonal
variation in foehn influence generally shows a maximum in spring and a minimum in summer. Except
for October and November, violin plots for most months present a unimodal "spindle-type," indicative
of a concentrated distribution. However, October and November uniquely display a bimodal pattern with
a secondary, weaker peak in the lower range, suggesting that, while the majority of foehn days in these
months affect over 50% of stations, a considerable portion (approximately 40%) still experiences a
limited foehn impact zone.

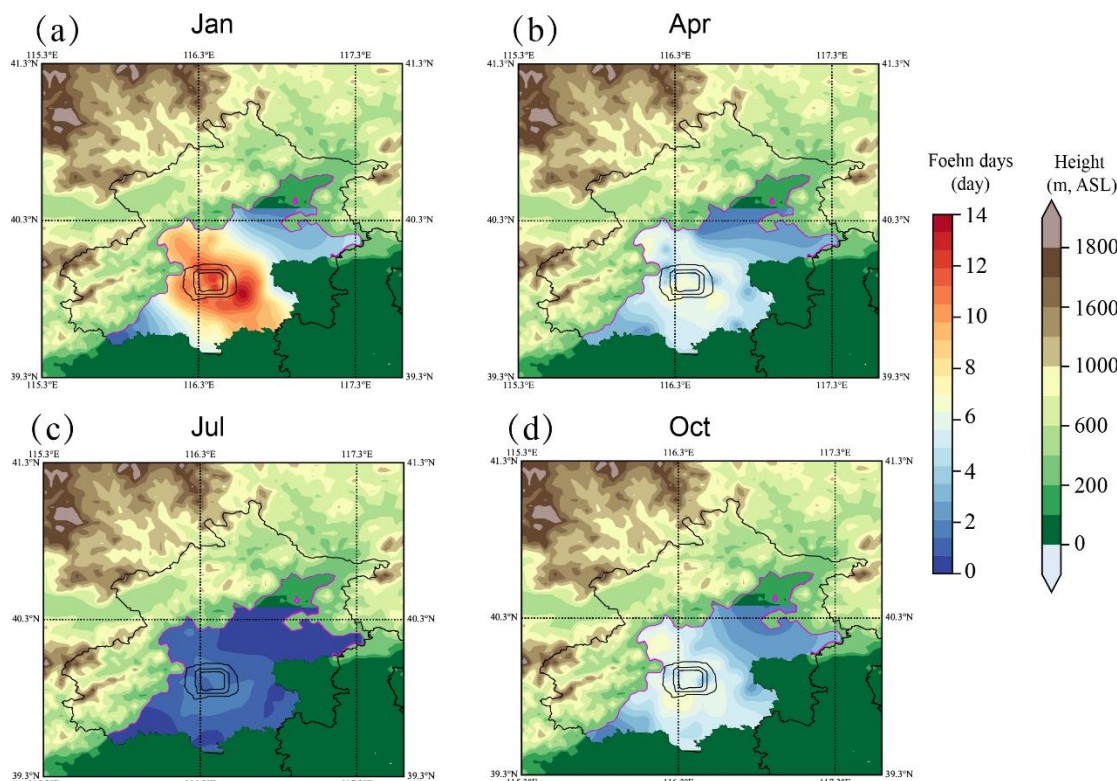


**Figure 5: Multi-year average monthly distribution of foehn days. The pink lines indicate the contour line at**

**an elevation of 200 m.**

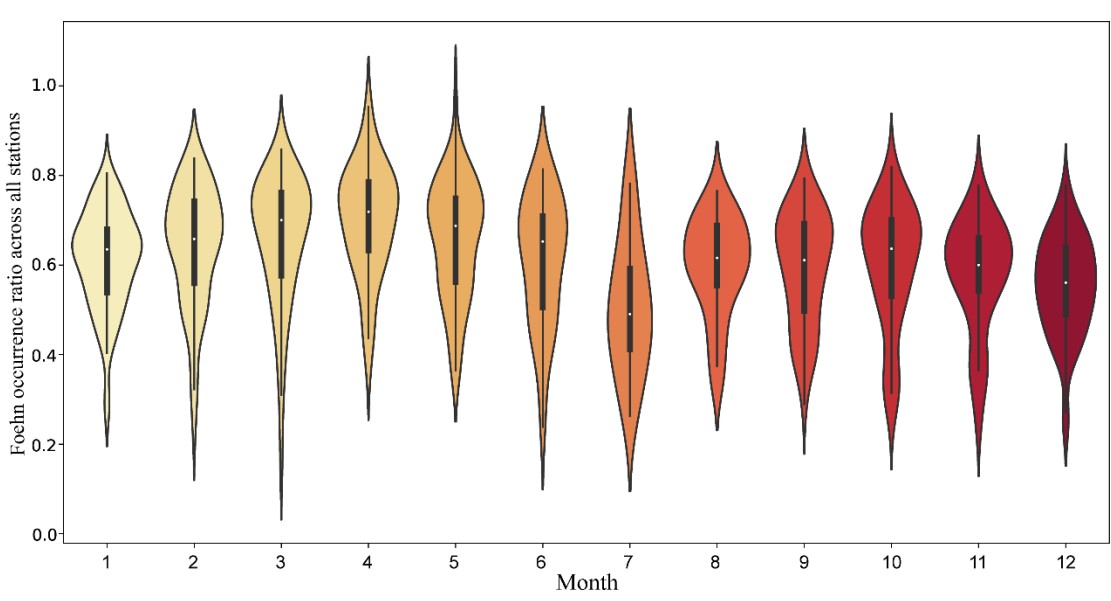

319

**Figure 6: Violin plots depicting the proportion of PAWSs experiencing foehn winds for each month.**

To assess the variations in temperature rise induced by foehn winds across different locations, we selected four national meteorological stations—Changping (CP), Haidian (HD), Chaoyang (CY), and Tongzhou (TZ)—situated along a path extending from the leeward side of the northwest mountains toward the southeastern plain (Fig. 1). We analyzed their hourly temperature difference ($\triangle$T), relative humidity difference ($\triangle$RH), and wind speed difference ($\triangle$WS) on foehn days. As shown in Table 3, the

median values of △T at these stations are highly consistent, ranging from 1.7–1.8 °C. The mean △T spans 2.0–2.2 °C, with the most pronounced increase observed at Station HD and the smallest at Station TZ. The maximum △T is greatest at TZ (11.8 °C), followed by HD (10.1 °C), and then CP (7.5 °C). When examining the 25th and 75th percentile values, half of the hourly warming instances at each station fall within a 1.3–2.6 °C range; however, the warming span for TZ is narrower than the other three stations, confined to 1.3–2.4 °C. The average values of △RH at each station all show a decrease, ranging from -8% to -11%. The maximum reduction of △RH reaches its highest value at HD (-75%) and the lowest at CP (-59%). The 25th and 75th percentiles of △RH fall between -14% and -3%. The average values of △WS at each station all show an increase, ranging from 0.4 to 0.7 m/s. The maximum value of the maximum △WS at each station is observed at TZ (8.8 m/s), and the minimum at HD (4.8 m/s). The 25th and 75th percentiles of △WS are between -0.3% and 1.4 m/s. The negative values of △WS may be related to the decrease in wind speed during consecutive foehn hours after the passage of the foehn.

**Table 3.** Summary statistics of hourly differences in temperature (△T, °C), relative humidity (△RH, %), and wind speed (△WS, m s$^{-1}$) at the four studied stations.

| | △T (℃) | | | | △RH (%) | | | | △WS (m s$^{-1}$) | | | |
| --- | --- | --- | --- | --- | --- | --- | --- | --- | --- | --- | --- | --- |
| | CP | HD | CY | TZ | CP | HD | CY | TZ | CP | HD | CY | TZ |
| **max** | 7.5 | 10.1 | 8.7 | 11.8 | -1 | -1 | -1 | -1 | 8.2 | 4.8 | 5.7 | 8.8 |
| **min** | 1.0 | 1.0 | 1.0 | 1.0 | -59 | -75 | -72 | -67 | -4.1 | -4.2 | -3.0 | -3.0 |
| **median** | 1.7 | 1.8 | 1.7 | 1.7 | -5 | -6 | -7 | -6 | 0.5 | 0.3 | 0.5 | 0.4 |
| **mean** | 2.1 | 2.2 | 2.1 | 2 | -8 | -9 | -11 | -10 | 0.7 | 0.4 | 0.6 | 0.6 |
| **25th Percentile** | 1.3 | 1.3 | 1.3 | 1.3 | -10 | -12 | -14 | -13 | -0.3 | -0.2 | -0.1 | -0.2 |
| **75th Percentile** | 2.6 | 2.6 | 2.5 | 2.4 | -3 | -3 | -4 | -3 | 1.4 | 1.0 | 1.2 | 1.1 |

In terms of daily variations in hourly temperature changes across these stations (Fig. 7a–d), the periods of minimal warming (troughs) typically occur from midday until before sunset. The trough for CP, which is nearest to the mountains, lasts from 11:00 AM to 8:00 PM, while for HD, CY, and TZ these periods are from 12:00 PM to 5:00 PM, 12:00 PM to 3:00 PM, and 12:00 PM to 5:00 PM, respectively. All stations exhibit two pronounced peaks in their hourly warming patterns each day, occurring around midnight before sunrise and around 8:00 or 9:00 AM post sunrise. There are also typically milder warming peaks around sunset, with HD displaying the most pronounced pre-sunset warming peak among the stations. Observing the monthly changes in hourly warming (Fig. 7e–h), CP, being closest to the mountains, exhibits the broadest range of warming fluctuations, whereas TZ, the farthest from the mountains, shows the narrowest range. The monthly mean warming values are typically lowest in July for all stations, coinciding with the month having the smallest range of warming fluctuations. The average peak warming values are generally seen during autumn (September to November), while the most substantial hourly warming spikes are noted in February.

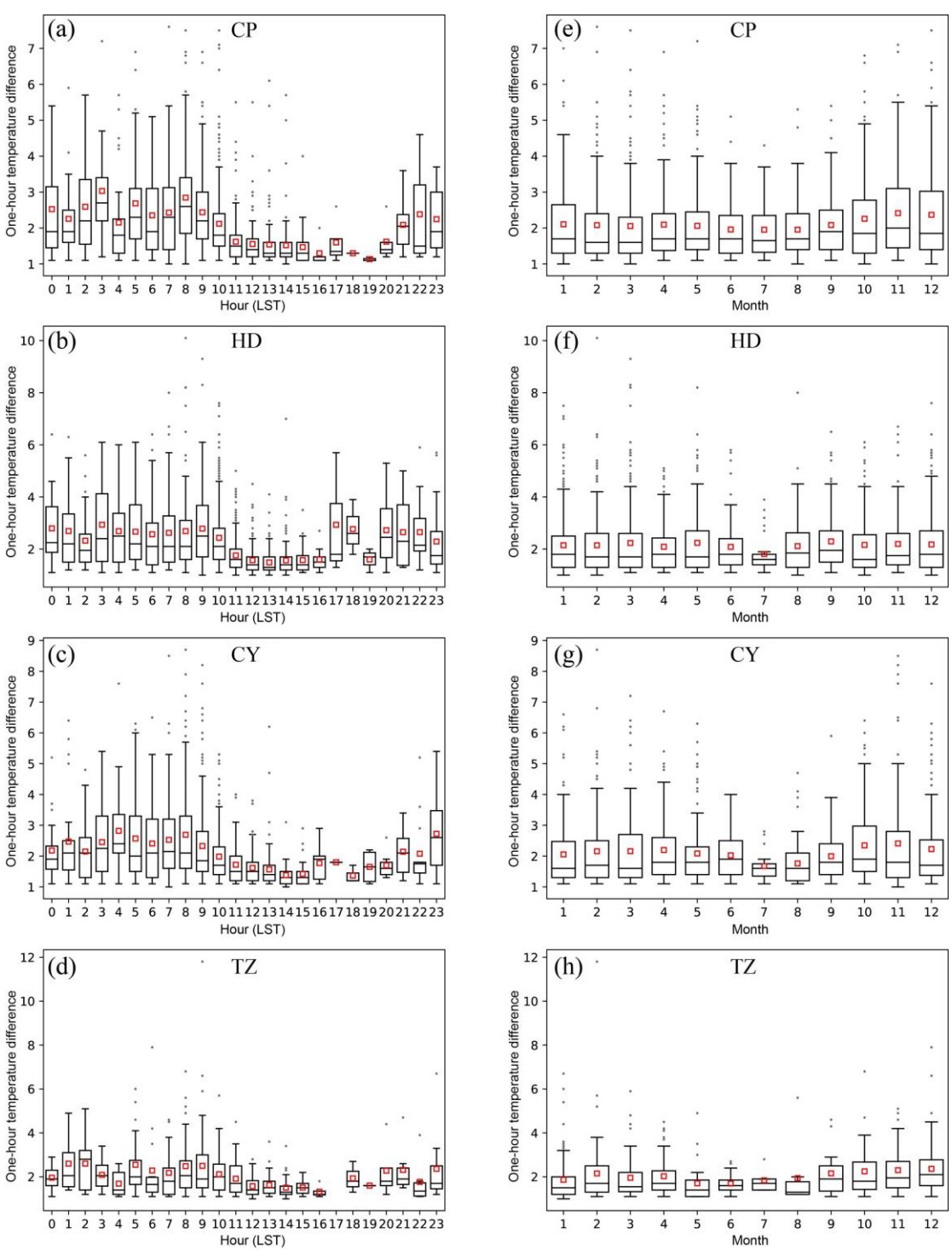

**Figure 7: Temporal variations in the hourly temperature changes: (a–d) diurnal variations and (e–h) monthly variations. Box-and-whisker plots show the 25th, 50th (median), and 75th percentiles, with the red square indicating the mean value.**

## 5. Relationship between pollution episodes and foehn events

In accordance with the definition of pollution episodes outlined in Chapter 2, 204 qualified pollution episodes during 2015–2020 were identified and visualized in Figure 8. Here, each pollution episode is

aligned according to its initiation time within the 0-23 Local Standard Time (LST) on the x-axis, with semi-transparent filled line plots illustrating PM2.5 concentration versus time. This alignment methodology facilitates the identification of LST-dependent PM2.5 variation characteristics through composite plotting. While the terminal positions of individual episodes generally correspond to their duration (in hours), it should be noted that these plotted durations may exceed actual episode lengths in most cases, though never surpassing 24 hours. The durations of these pollution episodes mostly remain under 4 days, with instances exceeding 7 days being rare. A conspicuous diurnal pattern in pollutant concentrations is evident, characterized by lower levels during the day and elevated concentrations at night (Fig. 8a). Figure 8b features gray bars that denote the sum of stations experiencing a foehn event at any given time (only considering the 14 plain national stations), reflecting the horizontal reach of the foehn winds. The timing of these peak occurrences aligns with the troughs of pollutant concentrations in Figure 8a, indicating that widespread foehn occurrences coincide with lulls in pollution concentrations. Red scatter points represent the maximum hourly temperature increases during foehn episodes for each time point, and their number for a given moment also signifies the number of pollution episodes experiencing foehn winds at that time. Evidently, foehn winds are more frequent during shorter pollution episodes; as the duration of a pollution episode extends, the likelihood of encountering foehn winds decreases. Generally, the maximum warming magnitude induced by foehn winds tends to decrease as the pollution episode persists longer. Statistics of the 204 pollution episode durations (in hours) reveal a median of 76.6 hours, a mean of 73 hours, a maximum of 313 hours, a minimum of 7 hours, and 25[th] and 75[th] percentiles of 42 and 101 hours, respectively. The proportion of foehn durations in all pollution episodes is depicted in Figure 9. Among the 204 pollution episodes, 67% (137 episodes) involved foehn occurrences. There is a negative correlation between the proportion of the foehn duration and the length of the pollution episodes, suggesting that longer-lasting pollution episodes see a lower proportion of time affected by foehn winds. On average, foehn winds account for 14.8% of pollution episode durations, reaching a maximum of 55.6% for episodes lasting 18 hours and plummeting to a minimum of 0.6% for episodes enduring 157 hours.

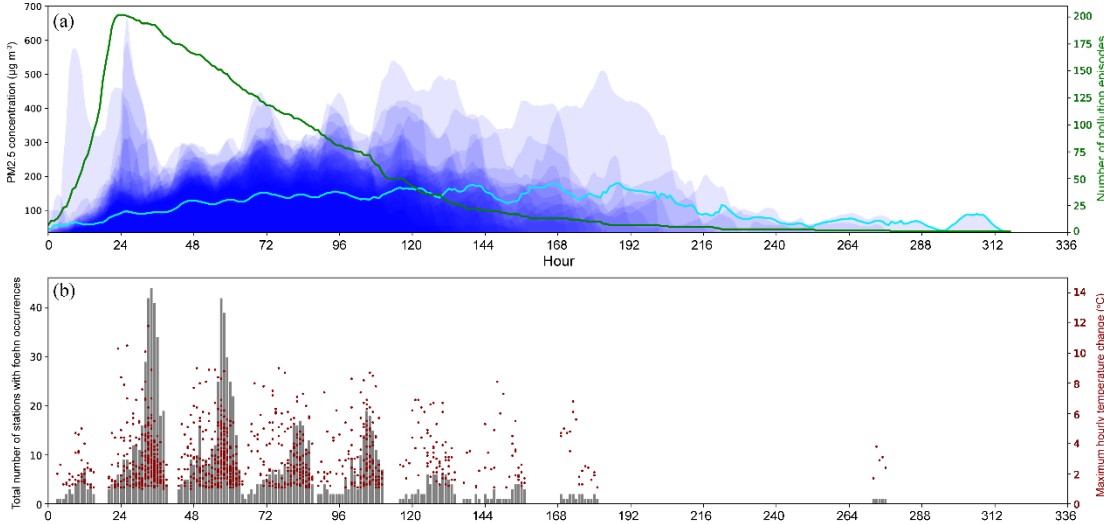

**Figure 8: Characteristics of pollution episodes and associated foehn occurrences. Each pollution episode is aligned according to its initiation time within 0–23 LST on the x - axis. (a) Temporal variation of the PM 2.5 concentration during pollution episodes, with the green line representing the number of pollution episodes and the light-blue line indicating the average PM 2.5 concentration of pollution episodes. (b) Foehn occurrence during pollution episodes, with gray bars indicating the cumulative number of stations with foehn occurrences**

**per episode (considering only the 14 plain national stations) and red scatter points representing the maximum hourly temperature change.**

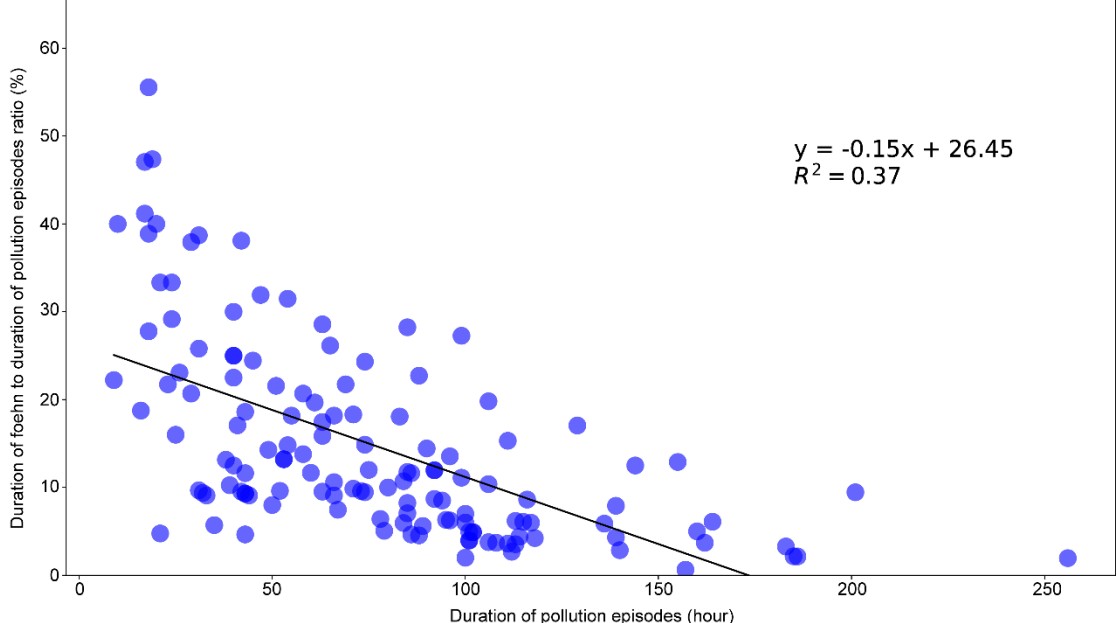

**Figure 9: Correlation between pollution episode duration and the foehn-to-pollution ratio.**

Figure 10a illustrates the relationship between hourly variations in PM2.5 concentrations and the number of sites experiencing foehn winds. Only the 14 national stations situated in the plain areas are considered for counting the foehn-affected sites. More often than not, foehn events are associated with a decline in the PM2.5 concentrations. For processes with a wider foehn influence (more sites reporting foehn winds), the reduction in PM2.5 concentrations tends to be more pronounced. Close to 60% of foehn events have an impact range restricted to no more than two stations, with instances of foehn winds affecting over half of the stations being relatively infrequent. Figure 10b relates the hourly changes in PM2.5 concentrations to the temperature increase at these sites during foehn events. During foehn periods, 60.4% of the time a drop in PM2.5 concentrations occurs, while the remainder of the time there is an increase in PM2.5 concentrations. The correlation between the maximum temperature rise and changes in the PM2.5 concentrations is weakly negative. Pronounced increases in the PM2.5 concentrations, such as hourly increments exceeding 50 μg/m³, mainly occur during mild warming phases with temperature increases of less than 2 °C.

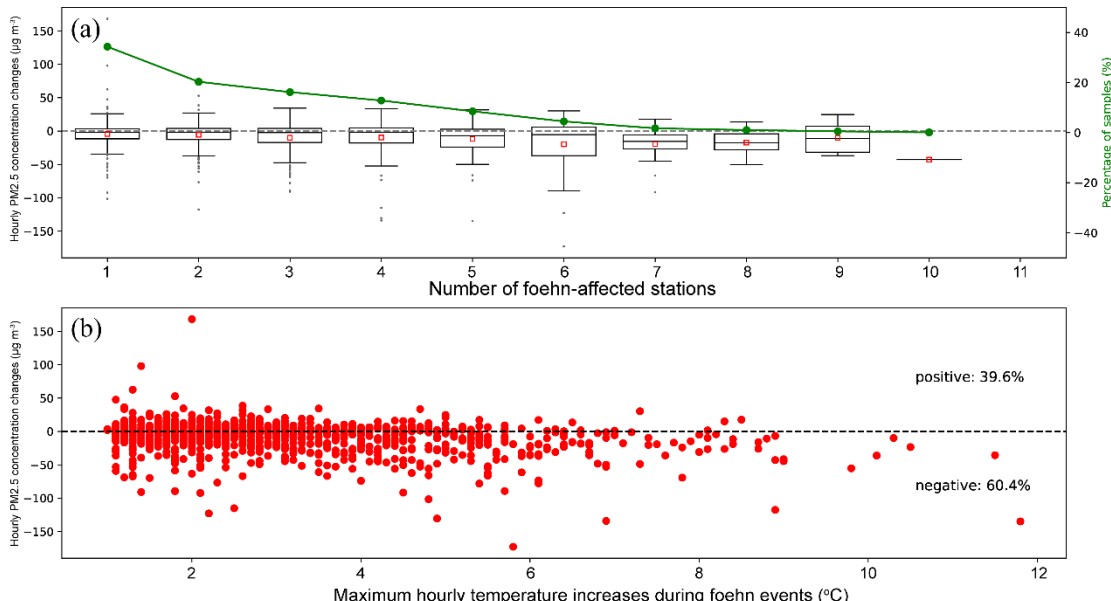

**Figure 10: Relationship between foehn events, PM2.5 concentration changes, and temperature variations. (a) Hourly PM2.5 concentration changes in relation to foehn occurrence. Box plot: distribution of hourly PM2.5 concentration changes. Green line: percentage of samples in each category. (b) Correlation between hourly PM2.5 concentration changes and maximum hourly temperature increases during foehn events.**

Figure 11 illustrates a typical pollution episode influenced by foehn events, occurring from January 6 to January 9, 2015. Foehn winds were observed during three distinct phases of this episode. Phase I: The foehn initially appeared at isolated stations at 04:00 on January 7, expanding to a widespread occurrence by 09:00. During this phase, pollutant concentrations exhibited a marked decrease, with the widespread foehn event closely following the trough in pollutant concentrations. Phase II: At 09:00 on January 8, a foehn was recorded at four national stations, after which the spatial extent of the foehn influence diminished. During the foehn-affected period, PM2.5 concentrations showed a slight decrease. However, this was followed by a rapid increase in PM2.5 levels, reaching a peak at 22:00. Phase III: Foehn winds reappeared at a single station at 00:00 on January 9, with their influence expanding after 01:00 and reaching maximum extent by 04:00. This phase corresponded to the pollutant clearance stage of the episode, characterized by a rapid decline in PM2.5 concentrations. This case study exemplifies the complex interactions between foehn winds and pollution dynamics, demonstrating both the potential for foehn events to facilitate pollutant dispersion and their role in subsequent rapid accumulation of pollutants under certain conditions.

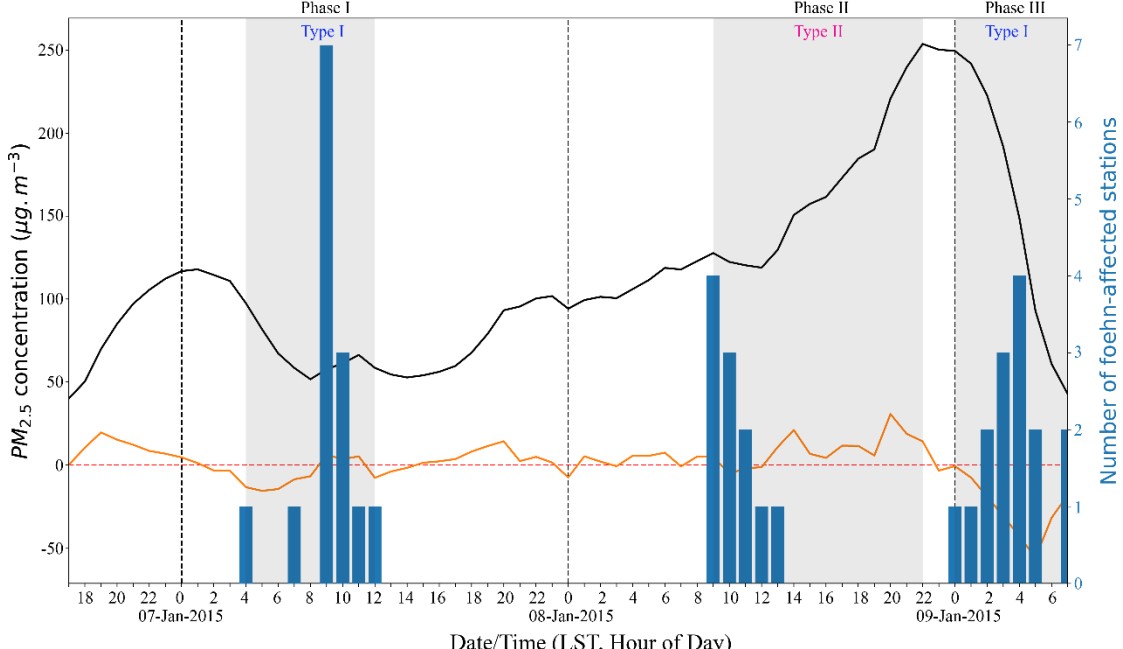

Figure 11. Temporal evolution of PM2.5 concentrations and foehn event occurrence during a pollution episode. Primary Y-axis (left): PM2.5 concentration (black line) and hourly PM2.5 concentration change (orange line). Secondary Y-axis (right): Number of stations experiencing foehn events (blue bars). Three distinct phases of the pollution episode are highlighted with gray-shaded areas, corresponding to different foehn-type classifications.

To systematically investigate the impact of foehn events on air pollution episodes, the following definitions were established: A foehn event is defined as a sequence of continuous or quasi-continuous foehn hours lasting at least 2 hours, where quasi-continuity allows intervals of up to 2 hours between successive foehn hours, subsequently merged into a single event. Foehn events were classified into two distinct types according to PM$_{2.5}$ concentration dynamics. Type I refers to rapid decline in pollutant concentration during foehn events, defined by the simultaneous satisfaction of two criteria: (1) for a foehn event, the PM$_{2.5}$ concentration change ($\Delta$C)—defined as the concentration at the hour immediately before event initiation minus the concentration at event termination—must be negative; (2) a 25% reduction in PM$_{2.5}$ concentration at the event termination compared to the initial value. Type II refers to rapid increases in pollutant concentrations following the termination of foehn events, defined by the combined fulfillment of the following two criteria: (1) a non-negative $\Delta$C, or a negative $\Delta$C with a terminal concentration reduction of less than 5% from the initial value; (2) the emergence of a new PM$_{2.5}$ peak with a concentration increase of more than 25% from the initial value, occurring either before subsequent foehn events if there are any or within 24 hours after this foehn event termination if there aren't.

All identified foehn events underwent rigorous screening across pollution episodes, complemented by manual validation to ensure methodological robustness. We classified the 204 pollution episodes involving foehn effects into these two categories, identifying specific dates corresponding to each type, comprising 80 days for Type I and 33 days for Type II. For compatibility of self-organizing map (SOM) analysis with daily ERA5 sea-level pressure (SLP) data, only days exhibiting single-type foehn events (exclusively Type I or II) were included. Employing the SOM methodology on ERA5 data for these categorized dates, we derived weather typing characteristics that differentiate the impacts of the two

foehn types on pollutants. For Type I (depicted in Fig. 12), a consistent high-pressure system is observed northwest of Beijing, accompanied by a pressure gradient directed from northwest to southeast. To quantify the pressure gradient, we use the pressure difference ($\triangle$P) between the center of the Beijing Plain (the ring road's center in Fig. 1) and a point 300 km northwest of this center. Notably, SOM types SOM2 and SOM4, which feature strong high-pressure systems and pronounced pressure gradients ($\triangle$P > 6 hPa), jointly account for 36.25% of occurrences. These conditions are frequently associated with the passage of cold fronts, facilitating the rapid dispersion of pollutants. SOM3 and SOM5 share similar pressure patterns to SOM2 and SOM4 but exhibit weaker pressure gradients (3 hPa < $\triangle$P < 6 hPa), collectively representing 30% of instances. The weakest pressure gradients ($\triangle$P ≈ 3 hPa) are observed in SOM1 and SOM6 types, together comprising 33.75% of cases. For Type II, Beijing is predominantly located within a near-isobaric field in SOM1 and SOM4, while SOM2 and SOM3 show weak pressure gradients ($\triangle$P ≈ 3 hPa) to the northwest or west of the city. Foehn winds under these types are generally weaker, resulting in only marginal decreases in pollutant concentrations. The subsequent rapid rise in pollutants could be attributed to boundary-layer processes induced by the foehn phenomenon, as suggested by Li et al. (2020).

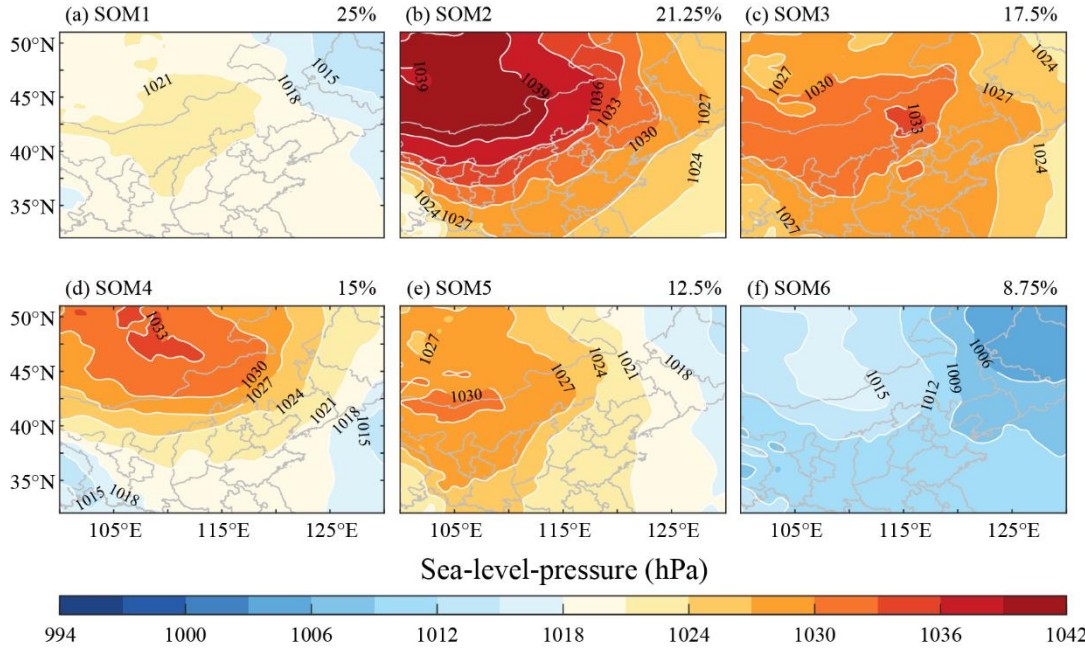

**Figure 12: Self-organized classification of the sea-level-pressure patterns associated with Type I foehn events.**

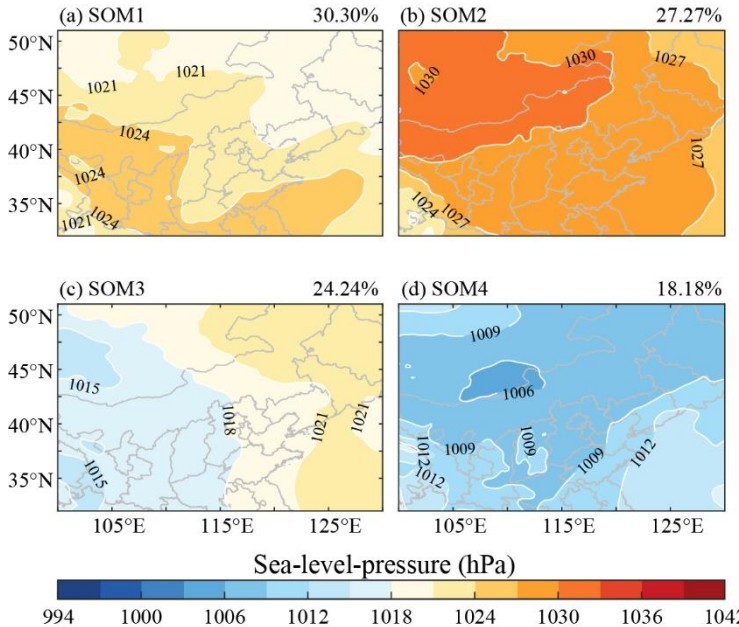

**Figure 13: Self-organized classification of the sea-level-pressure patterns associated with Type II foehn events.**

**6. Discussion**

The identification of foehn winds using AWS data requires careful differentiation from other warming mechanisms through characteristic meteorological signatures. Foehn events are distinguished by abrupt temperature increases (>1°C/hour), simultaneous humidity drops, and sustained winds aligned with mountain-plain airflow patterns (typically W/NW in Beijing), contrasting sharply with warm front-associated warming that exhibits gradual temperature rise, moisture increases, and E/SE winds. While tropical cyclone peripheral warming and anticyclonic subsidence demonstrate even lower thermal gradients and broader spatial impacts, our methodology employs wind-direction verification and thermal thresholds to effectively exclude these phenomena. Foehn warming differs significantly from solar radiation-induced warming in meteorological element change characteristics. Foehn warming is characterized by rapid short-term temperature surges accompanied by abrupt wind speed increases, sharp humidity drops, and clear directional movement from mountains to plains. In contrast, solar radiation warming lacks instantaneous abrupt changes in meteorological elements, with wind directions primarily influenced by local wind systems. In Beijing's plain areas without large-scale weather systems or foehn effects, mountain-valley and mountain-plain wind systems dominate, causing significant diurnal variations in near-surface wind directions: nocturnal winds blow from mountains to plains, while daytime winds reverse to blow from plains to mountains. At night without solar radiation, our foehn identification method effectively detects foehn at mountain-proximal stations due to the unidirectional mountain-to-plain airflow consistent with foehn movement. During daytime solar radiation warming without foehn, valley winds and plain-to-mountain winds cause plain station wind directions to point toward mountains (opposite to foehn direction), thereby preventing false foehn detection. However, downstream foehn propagation may lead to misidentification issues. Warming mechanisms at downstream stations involve a combination of foehn-related processes (advection and lee wave subsidence) and solar radiation heating (daytime only). Foehn advection may transport locally warmer air (e.g., urban heat island) downstream,

resulting in overestimation of foehn occurrence by our method. Daytime solar radiation exaggerates
foehn warming magnitude, while foehn-induced cloud-free or few-cloud conditions ("foehn clearance",
Hoinka, 1985a) further enhance solar radiation, creating a coupled direct-indirect foehn effect. Although
strictly meteorological criteria might classify such events as overestimations, the observed thermal
enhancements remain fundamentally tied to foehn-initiated processes, warranting their inclusion in
broader impact assessments of foehn phenomena.
Based on the preceding chapter's analysis, it emerges that the impact of foehn winds on pollution
events can be primarily categorized into two mechanisms: a reduction in pollutant concentrations and an
increase thereof, with the former accounting for over 60% of instances. These mechanisms are
respectively referred to as the direct and indirect effects of foehn winds on pollutants (Li et al., 2020).
The direct mechanism typically involves a strong pressure gradient perpendicular to the Taihang
Mountains, enhancing the intensity of the foehn (manifested by higher wind speeds and temperatures).
Northerly foehn winds often carry clean, cold air, leading to a rapid decline in pollutant concentrations
and even the termination of pollution episodes. This mechanism commonly operates during the terminal
phase of pollution events (the cleanup stage, as seen in Phase III of Fig. 11), though it may also occur in
the midst of pollution episodes if the foehn is not potent enough to fully dissipate pollutants (Phase I of
Fig. 11). In contrast, the indirect mechanism is more intricate. It corresponds to weather scenarios with
an isobaric field or weak pressure gradients. Under such mild meteorological settings, the region in front
of the mountains in Jing-Jin-Ji (Beijing-Tianjin-Hebei) is prone to developing local circulations
converging towards the mountain front, resulting in the accumulation of pollutants in these areas (Wang
and Zhang, 2020). Here, a foehn initially appears on the leeward side, with the formation of a fast-moving,
low-pollution, warm-dry air mass advancing southward, encountering a slow-moving, high-pollution,
cold-wet air mass from the south, potentially creating a haze front (Li et al., 2020). Given the weak nature
of the foehn—characterized by low wind velocities—it fails to induce a rapid decrease or removal of
pollutants. Instead, the weak pressure gradient between opposing air masses drives a "seesaw-like"
exchange of air masses. The advancing warm-dry air also stabilizes the lower atmosphere by reinforcing
temperature inversions above the cold air mass, in which overlying warm air increases the strength of
these inversions. This enhanced atmospheric stability traps pollutants near the surface, even as the slow
northward migration of the haze front continues to advect high-concentration pollutants from southern
regions into relatively cleaner northern areas. Observations from Li et al. (2020) reveal that this
interaction is accompanied by distinct boundary-layer evolutions: as the foehn develops, low-level
northerly winds strengthen alongside subsidence motions, leading to a temporary decrease in boundary-
layer aerosol concentrations; while as the foehn weakens and the haze front passes, these winds shift to
weaker southerly flows, accompanied by rapid increases in boundary-layer relative humidity and
pollutant concentrations—signals of enhanced moisture and pollution transport from southern sources.
This indirect pathway is not a simple dilution process but involves complex interactions between frontal
dynamics and boundary-layer stability. The weak foehn initially reduces surface pollutants, yet the
northward advancement of the haze front and the subsequent breakdown of vertical mixing ultimately
lead to pollutant accumulation. These observations from Li et al. (2020) provide support for how mild
foehn conditions, through modulating boundary-layer structure and air-mass interactions, contribute to
persistent pollution patterns in the region, as evident in the "initial decrease followed by rapid surge"
behavior during the second phase in Figure 11. In another scenario, when the foehn is weaker, its indirect
effects may only induce marginal reductions in pollutant concentrations within limited areas of the
northern region, while the citywide average pollutant concentrations exhibit a persistent increase rather
than a decline.
**7. Conclusion**
This study utilized data from Beijing's operational AWS network from 2015 to 2020, developing a foehn
identification method specifically tailored for the Beijing plain area based entirely on AWS data. The
method integrates considerations of the upper-air wind orientation relative to topography, meteorological
element variations during foehn passages, and the progressive propagation of foehn winds from leeward
slopes to downstream areas. Utilizing this approach, an initial comprehensive climatological analysis of
foehn events in Beijing was conducted, revealing that the annual average number of foehn days in the
region is 56.5, with notable differences in mean and maximum foehn days across years, exhibiting
fluctuating trends over time. Seasonally, foehn events occur most frequently in winter, followed by spring
and autumn and then, finally, summer. Spatial distribution patterns of foehn days show a consistent band-
like high-value zone extending from northwest to southeast, with low-value zones primarily in
northeastern plains of Beijing, though these patterns vary across seasons. The spatial extent of the
foehn influence was more pronounced in 2015 and 2016 compared to other years in the study period.
Seasonally, the foehn influence reached its maximum extent in the spring and was most limited
during summer months. Foehn-induced maximum hourly temperature increases can exceed 11 °C, with
peak warming typically occurring from nighttime to early morning, while the minimum temperature
changes are generally observed from noon to pre-sunset. Monthly analysis reveals that stations near
mountains experience the largest fluctuations in temperature increases, whereas plain stations farthest
from the mountains show the smallest variations. The average magnitude of the temperature
increase across all stations typically reaches its minimum in July, with a comparatively smaller range of
fluctuations relative to other months. Conversely, the maximum temperature increases generally occur in
autumn. The most substantial foehn-induced hourly temperature rises are often observed in February.
Foehn winds in Beijing have intimate ties with air-pollution episodes, with approximately 67% of
pollution episodes accompanied by a foehn. There exists a negative correlation between foehn duration
and pollution episode length, where longer pollution episodes encompass a smaller proportion of foehn
periods. During pollution events, foehn events predominantly coincide with declining PM2.5
concentrations; among pollution episodes featuring foehn winds, 60.4% see a decrease in PM2.5, while
39.6% observe an increase. The relationship between the maximum temperature rise during foehn events
and changes in PM2.5 concentrations is weakly negative. Instances of PM2.5 concentrations surging
over 50 μg m$^{-3}$ primarily coincide with weak foehn events characterized by temperature increases below
2 °C. Foehn events influence pollution episodes through two primary mechanisms: a direct
mechanism causing rapid pollutant decrease, and an indirect mechanism characterized by a rapid
increases in pollutant concentrations following the termination of foehn. The former typically involves a
strong pressure gradient perpendicular to the Taihang Mountains, linked with cold air outbreaks, enabling
efficient pollutant clearance due to stronger foehn winds; the latter occurs in milder meteorological
settings, with weak foehn winds only marginally lowering pollution levels, insufficient for clearance, and
subsequently, alterations to local flow fields and boundary-layer structures by foehn winds lead to rapid
pollutant accumulation and increases.
The foehn identification method proposed in this study, which relies solely on surface AWS data,
facilitates the identification of foehn events using long-term historical observational data. For
climatological studies of foehn winds worldwide, the application of methodologies analogous to those
presented herein enables the analysis of long-term observational datasets from a limited number of
surface meteorological monitoring stations. This approach facilitates a deeper understanding of how
foehn phenomena evolve and contribute to temperature increases in the context of global warming.
Additionally, it enhances researchers' ability to investigate the relationships between foehn winds and
high-impact weather phenomena, such as air pollution and heatwaves.

*Data availability.* The PM$_{2.5}$ data are available on the website https://quotsoft.net/air/. Other data can be requested from the corresponding author (jli@ium.cn).

*Author contributions.* JL had the original idea; JL, JZ, MB, JS, QL, and XJ performed the integrative data analysis; JL and MB wrote the manuscript. All authors discussed the results and commented on the paper.

*Competing interests.* The authors declare that they have no conflict of interest.

*Acknowledgments.* The authors would like to thank the anonymous reviewers for their helpful comments. This work was supported by the Beijing Natural Science Foundation (8222048), National Key R&D Program of China (2023YFC3007805), and the Open Grants of the State Key Laboratory of Severe Weather (2022LASW-A03). Thanks to the Beijing Meteorological Data Center for providing the observational data from AWSs.

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
