# Peer review of "Identification and characterization of foehn events in Beijing and their impact on air-pollution episodes"

_EGUsphere, 2024_

## Referee Comment (RC1)

This manuscript presents an innovative and practical approach to identifying foehn events using automatic weather station data, offering valuable insights into their climatology and impact on air pollution in Beijing. The strengths include the novel methodology, comprehensive spatial-temporal analysis, and the exploration of foehn-pollution interactions, which address a critical gap in the literature. However, the study could be enhanced by justifying key methodological choices, validating findings with additional data or techniques, and expanding the discussion of physical mechanisms. I have a few points that could be addressed to strengthen the manuscript and some minor comments.

General Comments:

1. The paper proposes a foehn identification method based solely on AWS data. While this approach is innovative, the authors should further clarify and justify the selection of criteria—for example, the choice of thresholds for temperature change, wind direction, and relative humidity. In several places, the rationale for setting specific ranges (e.g., wind directions "250–405°") is not fully explained. The upper bound exceeding 360° raises concerns about potential typographical or conceptual errors that need clarification. The authors should provide a detailed rationale explaining why this specific criterion was selected over alternatives (e.g., a fixed temperature threshold or a percentile-based approach). Additionally, a sensitivity analysis assessing how variations in this criterion affect foehn identification would strengthen the method's credibility and robustness.

The method uses temperature increases (>1°C per hour), humidity decreases, and specific wind directions to identify foehn events. However, these characteristics could also result from non-foehn processes, such as diurnal heating or synoptic-scale warm fronts, especially at downstream plain stations. The authors acknowledge potential overestimation at these sites (Section 6, lines 424–425), but more discussion or quantitative validation is needed to assess the method's specificity. How well does it distinguish foehn events from similar meteorological phenomena?

Besides, the use of the Self-Organizing Maps (SOM) technique is described, yet details on how the optimal number of nodes (six for Type I and four for Type II events) were determined are somewhat brief. A sensitivity analysis or additional justification would strengthen confidence in the clustering results.

3. I am not following the logic of relating the identified foehn events to air pollution. The authors filtered out high PM2.5 episodes first and then explored how the foehn events behave during these episodes. I think the proper way is the opposite by looking into how PM2.5 changes with and without foehn events. In addition, the study categorizes foehn impacts on PM2.5 into Type I (rapid decrease) and Type II (slight decrease followed by rapid increase), based on manual classification of 204 pollution episodes (Section 5, lines 386–389). This manual process may introduce subjectivity. Could an automated approach (e.g., clustering based on PM2.5 change rates and temperature increases) or statistical validation (e.g., significance testing of differences

between types) be applied to ensure objectivity and reproducibility? This would bolster confidence in the findings. The direct (strong pressure gradients) and indirect (weak gradients and boundary-layer changes) mechanisms linking foehn to pollution are well-described (Section 6, lines 429–451). However, the indirect mechanism—where weak foehn winds lead to pollutant accumulation via local circulations and inversions—could be elaborated with more evidence or modeling support. For instance, how does the "seesaw-like exchange" (line 446) manifest in wind fields or boundary-layer height data? Referencing specific observations from the case study (Figure 11) or prior work (e.g., Li et al., 2020) could clarify this process.

Minor Comments:

Lin 131-132: Pollution episodes are defined as periods with city-wide average PM2.5 >35 $\mu g\ m^{-3}$ and a mean >75 $\mu g\ m^{-3}$. This threshold aligns with practical air quality concerns but a brief justification or comparison with established criteria in the literature would contextualize this choice and enhance its applicability.

Line 136: I think it is ERA5. Better to include the version here. The NCEP data mentioned in Line 159 is a typo? Also, The reanalysis data at 0.25° × 0.25° resolution for weather pattern classification is appropriate for large-scale patterns but may miss fine-scale topographic effects critical to foehn dynamics in Beijing's complex terrain. Have the authors considered higher-resolution datasets (e.g., regional models) to verify small-scale features? Discussing this limitation would strengthen the methodology.

Figure 1: White triangles are hard to see. You may need to make them thicker.

Line 193: How were these 22 cases identified?

Figure 2: Is CE-PAWS included in Non-NM-PAW? The national AWS is used to identify the events first. Is it possible that the regional stations have foehn events while national stations do not so you are missing some event days?

Figure 3: The color scale should be the same to have an easy comparison. The band-like distribution of foehn days from the northwestern mountains to the southeastern plains is visually compelling, but the paper lacks a detailed physical explanation. Is this distribution driven by the Taihang Mountains' topography, prevailing northwest winds, or a combination of factors? Linking the spatial pattern to specific atmospheric or topographic mechanisms would enhance the analysis.

Line 242: Any explanations about what drives this narrowing trend?

Table 2: While this seasonal variation is well-documented, the underlying reasons are not explored. Are winter maxima linked to stronger pressure gradients, more frequent cold fronts, or topographic amplification of downslope winds? Including a brief discussion of potential meteorological drivers would provide deeper insight into the climatology of foehn events in Beijing.

Table 3: Can relative humidity and wind conditions be discussed in addition to temperature?

Figure 8: It is confusing how the figure is made and more explanations in the texts would aid comprehension. It says "each episode's initiation is marked by its Local Standard Time (LST)", but the axis has larger numbers than 24. Also, the subtitle of this figure is not very clear caused by the semi-colons. The axis labels are too small to read, which also applies to some other figures (such as Figure 9).

Line 490. The AWS-based foehn identification method is a key strength, enabling long-term analysis with widely available data. However, the paper does not discuss its potential application to other regions with similar topography (e.g., other parts of the North China Plain or mountain-adjacent cities globally). Addressing generalizability would increase the study's impact and relevance to the broader atmospheric science community.

---

## Referee Comment (RC2)

"Identification and characterization of foehn events in Beijing and their impact on air-pollution episodes."

The paper introduces a novel foehn identification method for the Beijing region, emphasizing seasonal and spatial patterns influenced by local topography and meteorological conditions. The potential strength of this work lies in the simplicity of the proposed method, which relies exclusively on near surface meteorological observations, making it suitable for long-term climatological applications. However, the authors do not clearly describe and discuss this methodological innovation throughout the manuscript. While the study presents a promising approach, the manuscript would benefit from improvements in clarity, methodological transparency, and deeper discussion of the results and their implications.

The study first identifies 204 pollution episodes in Beijing based on city-wide PM2.5 concentrations exceeding 75 μg/m³. A method is then applied to detect foehn events using near surface meteorological data, considering wind direction, temperature increase, humidity drops, and no precipitation. The authors cross-reference the pollution episodes with foehn occurrences and find that 137 of them overlap. These are manually classified into two types: Type I (rapid PM2.5 decrease) and Type II (slight decrease followed by a rapid increase). The classification appears to be subjective, based on visual inspection of time series. Finally, Self-Organizing Maps (SOMs) are used to identify distinct synoptic patterns associated with each foehn type.

**General comments:**

1)   The authors use PM2.5 time series during pollution episodes (defined by concentrations exceeding 75 μg/m³) to classify events into Type I and Type II. However, it is only between lines 382 and 401, in reference to Figure 11, that we get a clearer idea of what is meant by the "rapid pollutant concentration decreases" (Type I) and a "slight pollutant concentration decreases followed by a rapid increase" (Type II). Based on Figure 11, a "rapid" decrease seems to occur over approximately 6 hours, is that correct? I assume not all cases follow the same time window. Could the authors clarify how this visual classification was performed? Was any threshold defined? This point is particularly relevant if the method is to be applied to longer time series, where visual inspection alone may not be feasible.

2)   When the authors mention that the foehn identification method was developed based on 22 representative historical cases, it is not clear whether these cases were derived from previous literature or from the same dataset used in this study. It would be helpful to clarify this point more explicitly. Were these 22 historical cases associated with pollution episodes? Were they classified as Type I, Type II or both?

3) I appreciate the use of standard meteorological data from AWS to identify foehn patterns, and this strength could be further emphasized throughout the text. While I value the simplicity of the approach, I wonder whether the authors have access to eddy covariance system data to quantify

turbulence-related variables, such as turbulent kinetic energy (TKE) or the standard deviation of vertical velocity. These metrics could provide direct evidence of turbulence intensity and offer a more detailed view of how foehn winds enhance vertical mixing and potentially contribute to pollutant dispersion.

**Minor comments:**

-The captions of several figures are unclear and lack essential details. I recommend that the authors carefully review each figure and revise the captions to provide complete and self-explanatory information.

**Line 231**: the caption of Figure 3 states: "Annual distribution of foehn days*",* but the accompanying map appears to include a Digital Elevation Model (DEM). However, there is no legend to explain the meaning of the colors used.

**Line 276:** A similar issue applies to Figure 5; the color shading is not accompanied by a legend.

**Line 308:** In Figure 7, what does the red square represent?

**Line 342**. The linear regression equation and axis labels are too small and difficult to read.

**Line 378:** Figure 11 illustrates a case study showing the temporal evolution of PM2.5 concentrations and foehn event occurrences during a pollution episode. While the authors refer to "phases" (Phase I, Phase II, and Phase III), these phases are not marked in the figure. The dashed vertical lines might correspond to these phases, but this is not explicitly stated. The figure caption lacks clarity and does not adequately guide the reader.

**Line 398**: The term "weak pressure gradients" is used, but no numerical values are provided (e.g., 1–3 hPa over 500 km?).

**Line 403:** Throughout the manuscript, the authors frequently omit units, symbols. For example, in Figure 12, the unit of sea level pressure is not indicated.

**Line 411 – 416**: The authors acknowledge the challenge of distinguishing foehn-induced warming from other non-foehn processes, such as solar radiation or warm air advection. While they suggest that this issue could be mitigated by incorporating detailed analyses of wind fluctuations and upstream–downstream consistency (in Zhang and Li, 2024), such an approach is not actually applied in the current study. As a result, the risk of misclassifying non-foehn warming as foehn, especially at downstream locations, remains unresolved. Explicitly addressing this limitation, or testing the proposed checks within the study itself, would greatly improve the credibility and robustness of the results.

---

## Author Response (AR1)

**Response to reviewer's comments**

We sincerely thank the reviewer for in-depth comments and helpful suggestions. Your feedbacks have been of great assistance in improving the quality and clarity of our work. We have responded to all the comments point-by-point and made corresponding changes in the manuscript. Following are detailed responses to all the comments.

**Reviewer: 1**

This manuscript presents an innovative and practical approach to identifying foehn events using automatic weather station data, offering valuable insights into their climatology and impact on air pollution in Beijing. The strengths include the novel methodology, comprehensive spatial-temporal analysis, and the exploration of foehn-pollution interactions, which address a critical gap in the literature. However, the study could be enhanced by justifying key methodological choices, validating findings with additional data or techniques, and expanding the discussion of physical mechanisms. I have a few points that could be addressed to strengthen the manuscript and some minor comments.

General Comments:

1. The paper proposes a foehn identification method based solely on AWS data. While this approach is innovative, the authors should further clarify and justify the selection of criteria—for example, the choice of thresholds for temperature change, wind direction, and relative humidity. In several places, the rationale for setting specific ranges (e.g., wind directions "250–405°") is not fully explained. The upper bound exceeding 360° raises concerns about potential typographical or conceptual errors that need clarification. The authors should provide a detailed rationale explaining why this specific criterion was selected over alternatives (e.g., a fixed temperature threshold or a percentile-based approach). Additionally, a sensitivity analysis assessing how variations in this criterion affect foehn identification would strengthen the method's credibility and robustness.

Thank you for the comment.

Regarding the wind direction range originally expressed as "250–405°", we have corrected this to 250–360° or 0–45° (revised in Figure 2 and Line 200) to avoid circular ambiguity (405° exceeding 360°). This wind direction range setting is consistent with the topography of the major mountain ranges northwest of Beijing (as shown in the revised Figure 1, which now labels the mountain ranges). Foehn winds are dry, warm downslope winds generated by subsidence on the leeward side of mountains, requiring upstream airflow to intersect with mountain barriers at a sufficient angle to induce lee-side descent rather than parallel flow. The 250–360° range corresponds to airflow traversing the Taihang Mountains (northern section), Xishan Mountains, and Jundu Mountains, ensuring an intersection angle between wind direction and mountain orientation. The 0–45° range targets northeasterly winds interacting specifically with the Jundu Mountains. This wind direction criterion reflects the geographic constraints and airflow dynamics critical for foehn formation.

[Figure]

Figure 1: Distribution of observation sites in Beijing, China.

In identifying foehn events, we adopted the generalized definition of foehn: "wind warmed and dried by descent, in general on the lee side of a mountain" (WMO, 1992). As shown in Figure 2 of the original manuscript, the identification of foehn relies on the Near-mountain plain national AWS (NM-PNAWS). These NM-PNAWSs provide over 60 years of continuous observational data, comply with WMO siting and operational standards for surface meteorological observations, and adhere to strict data quality requirements. In contrast, regional AWS stations were mostly established in phases after 2000, and the majority do not meet WMO standards for site selection or observational guidance. Therefore, we selected high-quality, long-term NM-PNAWS data as the basis for determining foehn events and foehn days, ensuring a robust foundation for future climatological analyses of foehn.

NM-PNAWSs are located near mountain ranges and are the first to be affected by foehn winds. We initially attribute all dry, warm winds descending from mountains to foehn-related processes, then apply a temperature increase threshold to select moments with significant foehn impacts. During nighttime, the absence of solar radiation enables more accurate identification of foehn wind compared to daytime. Therefore, we focus on the impact of temperature thresholds on nighttime foehn wind by using the criterion that requires the top 15% of high-temperature-increase data (i.e., data in the ≥85th percentile) to determine the 1-hour temperature change threshold conditions. Both excessively high and low temperature increase thresholds can affect the proportion of selected foehn wind data. Taking Haidian (HD) Station as an example, a 1℃ per hour temperature increase threshold successfully selects the top 15% of high-temperature-increase data for all nighttime periods. Raising the threshold to 1.5℃ leads to failure in meeting

the top 15% requirement for some nighttime periods, while lowering it to 0.5℃ includes excessive weak warming data despite satisfying the requirement for all nighttime periods (Fig R1 and R2). Similar results were observed at other NM-PNAWS stations, such as Changping (CP) Station (Figs. R3 and R4). Based on this, we empirically established a criterion of selecting no less than the top 15% of high-temperature-increase data over 24 hours for foehn wind screening, corresponding to a 1℃ per hour temperature increase threshold. To maintain consistency, the same threshold was applied to non-near-mountain plain AWS stations (Non-NM-PAWS). Although foehn wind causes decreases in relative humidity (RH), RH change is unsuitable as a screening threshold; thus, our criteria only require a negative 1-hour RH change. Based on the preceding discussion, we have incorporated further elaboration on foehn wind screening criteria in the manuscript as follows: "*Foehn winds are dry, warm downslope winds generated by subsidence on the leeward side of mountains, requiring upstream airflow to intersect with mountain barriers at a sufficient angle to induce lee-side descent rather than parallel flow. The 250–360° range corresponds to airflow traversing the Taihang Mountains, Xishan Mountains, and Jundu Mountains (Fig. 1), ensuring an intersection angle between wind direction and mountain orientation. The 0–45° range targets northeasterly winds interacting specifically with the Jundu Mountains. By adopting a 1-hour warming threshold >1℃ (consistent with prior criteria), NM-PNAWSs can consistently capture the top 15% (≥85th percentile) of high-warming data over 24 hours. Increasing the threshold above 1℃ reduces data inclusion—notably at night—whereas decreasing it below 1℃ introduces excessive weak warming events.*"

[Figure]

**Figure R1: Cumulative probability plot of one-hour temperature difference at HD station**

[Figure]

**Figure R2: Violin plot of one-hour temperature increase from broadly defined foehn at HD station**

[Figure]

**Figure R3: Cumulative probability plot of one-hour temperature difference at CP station**

[Figure]

**Figure R4: Violin plot of one-hour temperature increase from broadly defined foehn at CP station**

The method uses temperature increases (>1°C per hour), humidity decreases, and specific wind directions to identify foehn events. However, these characteristics could also result from non-foehn processes, such as diurnal heating or synoptic-scale warm fronts, especially at downstream plain stations. The authors acknowledge potential overestimation at these sites (Section 6, lines 424–425), but more discussion or quantitative validation is needed to assess the method's specificity. How well does it distinguish foehn events from similar meteorological phenomena? Thank you for the comment. In large-scale weather systems, warm fronts, tropical cyclones, and anticyclonic highs can all cause surface temperature increases, but these warmings differ significantly from foehn-induced warming. Warm front warming typically features long duration but slow warming rate (typically <0.5°C/hour), accompanied by increased water vapor content. Warm fronts affecting Beijing mainly originate from the southeast, corresponding to dominant surface wind directions of east or southeast. In contrast, foehn warming is characterized by rapid short-term temperature increase with abrupt humidity drop, primarily arriving from mountainous areas in the west and northwest, corresponding to dominant surface wind directions of west or northwest. Based on these differences, the overestimation of warming at downstream plain stations discussed in this paper is unrelated to the warm front warming mechanism.

Although peripheral warming from tropical cyclones occurs extremely rarely in Beijing, it requires further differentiation from foehn warming. This type of warming originates from subsidence associated with divergent upper-level outflow of tropical cyclones, featuring low warming rate (≤0.1°C/hour, Zhang et al., 2010), wide but weak intensity. Our foehn identification methods can effectively filter out such warming through threshold setting.

Anticyclonic high warming results from weak subsidence (subsidence velocity ~10cm/s), characterized by slow warming rate and limited influence range. Its contribution to surface warming is significantly weaker than foehn and does not trigger foehn identification criteria.

Foehn warming differs significantly from solar radiation-induced warming in meteorological element change characteristics: foehn warming exhibits rapid short-term temperature surges with abrupt wind speed increases and humidity drops, with clear directional movement from mountains to plains. Solar radiation warming lacks instantaneous abrupt changes in meteorological elements, with wind directions mainly influenced by local wind systems. In Beijing plain areas without large-scale weather systems and foehn effects, mountain-valley and mountain-plain wind systems dominate, causing significant diurnal variations in near-surface wind directions: nocturnal winds blow from mountains to plains, while daytime winds reverse. At night without solar radiation, our foehn identification method effectively detects foehn at mountain-proximal stations. During daytime solar radiation warming without foehn, valley winds and plain-to-mountain winds result in plain station wind directions pointing toward mountains (opposite to foehn direction), preventing false foehn detection. However, foehn propagation downstream may cause misidentification. When foehn propagates from leeward plain stations downstream, observed warming at downstream stations includes both foehn-related warming (advection and lee wave subsidence) and solar radiation heating (daytime only). Foehn advection may transport local warm air (e.g., urban heat island) downstream, leading to overestimation of foehn occurrence by our method. Meanwhile, daytime solar radiation exaggerates foehn warming magnitude. Foehn-induced cloud-free or few-cloud conditions ("foehn clearance", Hoinka, 1985a) enhance solar radiation and contribute to warming, which can be broadly considered a combined direct and indirect foehn effect. Strictly speaking, foehn occurrence at downstream stations may be overestimated, but from a broader foehn impact perspective, these overestimations relate to foehn propagation.

Based on the preceding analysis, we have correspondingly revised the content in the discussion section of Chapter 6 as follows: "*The identification of foehn winds using AWS data requires careful differentiation from other warming mechanisms through characteristic meteorological signatures. Foehn events are distinguished by abrupt temperature increases (>1°C/hour), simultaneous humidity drops, and sustained winds aligned with mountain-plain airflow patterns (typically W/NW in Beijing), contrasting sharply with warm front-associated warming that exhibits gradual temperature rise, moisture increases, and E/SE winds. While tropical cyclone peripheral warming and anticyclonic subsidence demonstrate even lower thermal gradients and broader spatial impacts, our methodology employs wind-direction verification and thermal thresholds to effectively exclude these phenomena. Foehn warming differs significantly from solar radiation-induced warming in meteorological element change characteristics. Foehn warming is characterized by rapid short-term temperature surges accompanied by abrupt wind speed increases, sharp humidity drops, and clear directional movement from mountains to plains. In contrast, solar radiation warming lacks instantaneous abrupt changes in meteorological elements, with wind directions primarily influenced by local wind systems. In Beijing's plain areas without large-scale weather systems or foehn effects, mountain-valley and mountain-plain wind systems dominate, causing significant diurnal variations in near-surface wind directions: nocturnal winds blow from mountains to plains, while daytime winds reverse to blow from plains to mountains. At night without solar radiation, our foehn identification method effectively detects foehn at mountain-proximal stations due to the unidirectional mountain-to-plain airflow consistent with foehn movement. During daytime solar radiation warming without foehn, valley winds and plain-to-mountain winds cause plain station wind directions to point toward mountains (opposite to foehn direction), thereby preventing*

*false foehn detection. However, downstream foehn propagation may lead to misidentification issues. Warming mechanisms at downstream stations involve a combination of foehn-related processes (advection and lee wave subsidence) and solar radiation heating (daytime only). Foehn advection may transport locally warmer air (e.g., urban heat island) downstream, resulting in overestimation of foehn occurrence by our method. Daytime solar radiation exaggerates foehn warming magnitude, while foehn-induced cloud-free or few-cloud conditions ("foehn clearance", Hoinka, 1985a) further enhance solar radiation, creating a coupled direct-indirect foehn effect. Although strictly meteorological criteria might classify such events as overestimations, the observed thermal enhancements remain fundamentally tied to foehn-initiated processes, warranting their inclusion in broader impact assessments of foehn phenomena.*"

Besides, the use of the Self-Organizing Maps (SOM) technique is described, yet details on how the optimal number of nodes (six for Type I and four for Type II events) were determined are somewhat brief. A sensitivity analysis or additional justification would strengthen confidence in the clustering results.

Thank you for the comment.

The identification of the optimal number of nodes is primarily based on the following aspects. The first is that we prioritized minimizing quantization error while ensuring interpretability of patterns. Preliminary tests with node numbers ranging from 3–8 revealed that beyond 6 nodes for Type I or 4 nodes for Type II, clusters exhibited excessive fragmentation without meaningful meteorological distinctions. The second is that only 6 nodes for Type I and 4 nodes for Type II cleanly separated distinct SLP configurations (e.g., changes in trough/ridge) confirmed by expert evaluation. Based on the abovementioned two aspects, we have determined the optimal number of nodes for both Type I and Type II. Meanwhile, we have also supplied the relevant content in the revised manuscript. The specific contents have been rewritten in the revised manuscript as follows: "*The determination of the optimal number of nodes is based on two main considerations. First, we focused on minimizing quantization error while maintaining the interpretability of the identified patterns. Preliminary tests with node numbers ranging from 3 to 8 showed that when the node count exceeded 6 for Type I or 4 for Type II, the resulting clusters became overly fragmented, lacking meaningful meteorological distinctions. Second, expert evaluation confirmed that only 6 nodes for Type I and 4 nodes for Type II effectively separated distinct SLP configurations, such as changes in troughs and ridges. These configurations provided a clear and interpretable distinction between meteorological patterns. Based on these two factors—quantization error minimization and expert validation—we concluded that 6 nodes for Type I and 4 nodes for Type II are optimal for accurately capturing the relevant meteorological features.*"

3. I am not following the logic of relating the identified foehn events to air pollution. The authors filtered out high PM2.5 episodes first and then explored how the foehn events behave during these episodes. I think the proper way is the opposite by looking into how PM2.5 changes with and without foehn events.

Thank you for the comment.

The suggested approach of comparing PM2.5 variations under foehn and non-foehn conditions could indeed reveal their general correlation. However, our study specifically focuses on elucidating the mechanistic role of foehn events within pollution episodes rather than broadly

assessing their air quality impacts. The adopted methodological framework of first identifying pollution episodes then examining foehn characteristics during these episodes allows systematic investigation of 1) how foehn interacts with different pollution stages, and 2) its potential dynamic mechanisms in pollution evolution.

To address this concern, we have included a comparative analysis through boxplots of city-wide hourly PM2.5 concentrations under both conditions (Figure R5). The statistical characteristics demonstrate remarkable similarity between foehn and non-foehn scenarios, with both showing substantial outliers corresponding to high pollution events. This observation suggests that binary categorization based solely on foehn presence/absence cannot sufficiently disentangle its complex impacts on pollution processes.

Our process-oriented approach enables clearer identification of two distinct mechanistic pathways through which foehn events influence pollution evolution (Figure 10b). Subsequent process analysis (Figures 11-12) reveals how these mechanisms potentially interact with meteorological and emission factors to shape pollution dynamics.

[Figure]

**Figure R5: Boxplots of hourly city-wide average PM2.5 concentrations under foehn and non-foehn conditions. Foehn hours were identified using the algorithm described in the manuscript applied to observational data from NM-PNAWSs.**

In addition, the study categorizes foehn impacts on PM2.5 into Type I (rapid decrease) and Type II (slight decrease followed by rapid increase), based on manual classification of 204 pollution episodes (Section 5, lines 386–389). This manual process may introduce subjectivity. Could an automated approach (e.g., clustering based on PM2.5 change rates and temperature increases) or statistical validation (e.g., significance testing of differences between types) be applied to ensure objectivity and reproducibility? This would bolster confidence in the findings.

Thank you for the comment. Based on the subjective classification results, we established the following objective classification method and ensured the consistency between the subjective and objective classification results.

A foehn event is defined as a sequence of continuous or quasi-continuous foehn hours lasting at least 2 hours, where quasi-continuity allows intervals of up to 2 hours between successive foehn hours, subsequently merged into a single event.

Foehn events were classified into two distinct types according to $PM_{2.5}$ concentration dynamics. Type I: the type with rapid decline in pollutant concentration, which is defined by the simultaneous satisfaction of two criteria:

(1) for a foehn event, the $PM_{2.5}$ concentration change ($\Delta C$)—defined as the concentration at the hour immediately before event initiation minus the concentration at event termination—must be negative;

(2) a 25% reduction in $PM_{2.5}$ concentration at the event termination compared to the initial value.

Type II: the type with rapid increase in pollutant concentration or slight decrease first and then rapid increase, which is defined by the simultaneous satisfaction of two criteria:

(1) a non-negative $\Delta C$, or a negative $\Delta C$ with a terminal concentration reduction of less than 5% from the initial value;

(2) the emergence of a new $PM_{2.5}$ peak with a concentration increase of more than 25% from the initial value, occurring either before subsequent foehn events if there are any or within 24 hours after this foehn event termination if there aren't.

All identified foehn events underwent rigorous screening across pollution episodes, complemented by manual validation to ensure methodological robustness. For compatibility of self-organizing map (SOM) analysis with daily ERA5 sea-level pressure (SLP) data, only days exhibiting single-type foehn events (exclusively Type I or II) were included.

We have made expansions and revisions to the main text, and the revised content is: "*To systematically investigate the impact of foehn events on air pollution episodes, the following definitions were established: A foehn event is defined as a sequence of continuous or quasi-continuous foehn hours lasting at least 2 hours, where quasi-continuity allows intervals of up to 2 hours between successive foehn hours, subsequently merged into a single event. Foehn events were classified into two distinct types according to $PM_{2.5}$ concentration dynamics. Type I refers to rapid decline in pollutant concentration during foehn events, defined by the simultaneous satisfaction of two criteria: (1) for a foehn event, the $PM_{2.5}$ concentration change ($\Delta C$)—defined as the concentration at the hour immediately before event initiation minus the concentration at event termination—must be negative; (2) a 25% reduction in $PM_{2.5}$ concentration at the event termination compared to the initial value. Type II refers to rapid increases in pollutant concentrations following the termination of foehn events, defined by the combined fulfillment of the following two criteria: (1) a non-negative $\Delta C$, or a negative $\Delta C$ with a terminal concentration reduction of less than 5% from the initial value; (2) the emergence of a new $PM_{2.5}$ peak with a concentration increase of more than 25% from the initial value, occurring either before subsequent foehn events if there are any or within 24 hours after this foehn event termination if there aren't.*"

The direct (strong pressure gradients) and indirect (weak gradients and boundary-layer changes) mechanisms linking foehn to pollution are well-described (Section 6, lines 429–451). However, the indirect mechanism—where weak foehn winds lead to pollutant accumulation via local circulations and inversions—could be elaborated with more evidence or modeling support. For instance, how does the "seesaw-like exchange" (line 446) manifest in wind fields or boundary-layer height data? Referencing specific observations from the case study (Figure 11) or prior work (e.g., Li et al., 2020) could clarify this process.

Thank you for the comment.

To clarify the indirect mechanism, we reference Figure 1a, Figure 2a (redrawn from Li et al., 2020), Figure 9, and Figure 10 from Li et al., 2020 (attached below for convenience).

CP, AOT, and YZ in Figure 2 represent air quality monitoring stations located in the northern, central, and southern regions, respectively, with their positions shown in Figure 1. Figures 9 and 10 present observations from the IAP site, corresponding to Doppler lidar and a 325-meter meteorological tower, respectively. AOT is the air quality monitoring station closest to IAP.

Figure 9d, the signal-to-noise ratio (CNR) from Doppler lidar observations, shows the evolution of the boundary-layer structure throughout December 24, 2025. Before 05:00, aerosols accumulated near the surface, forming a shallow aerosol layer, while a thicker elevated aerosol layer existed aloft. Over time, the surface aerosol layer weakened significantly, and the elevated aerosol layer thinned and descended gradually. During this period, weak northerly winds dominated below several hundred meters in the boundary layer (Figure 9c). After sunrise, as the mixed layer developed, the vertical distribution of aerosols exhibited typical mixed-layer characteristics (after 10:00). CP was affected by foehn earlier, with its PM2.5 concentration dropping significantly at 12:00. Foehn reached IAP around 13:00 (Figure 10), causing a sharp decline in PM2.5 concentration at the AOT station, while the PM2.5 concentration at the southern YZ station remained at a peak (~ 500 μg/m³), creating a striking north-south gradient in pollutant distribution. During foehn, influenced by lee waves, the atmosphere showed obvious vertical fluctuations (Figures 9a and 10c). At 13:00, IAP experienced significant downward air motion (Figure 9a) and increased low-level northerly winds. Foehn led to a notable overall decrease in aerosol concentration within the boundary layer (Figure 9d). The low-pollution, warm-dry air mass formed by foehn advanced southward and confronted a slow-moving, high-pollution, cool-moist air mass from the south, forming a haze front. The position of the smog front depended on the balance between the two opposing air masses. Due to the weak foehn, its southward advance was limited, and the foehn wind gradually weakened over time. Under the weak pressure gradients between the air masses, the haze front moved slowly northward in a seesaw-like manner. Meanwhile, the interaction between warm and cold air masses caused the warm air to rise over the cold air, strengthening the inversion above the cold air and stabilizing the lower atmosphere, which further exacerbated surface pollution concentrations. Around 20:30, the haze front began to affect IAP, causing low-level winds to shift from northwesterly to weaker southwesterly (Figure 9), with rapid increases in boundary-layer relative humidity and pollutant concentrations (Figures 10b, 9d, and 2). As shown, weak foehn winds play a critical role in the northward propagation of southern pollutants.

[Figure]

Figure 1. (a) Locations of observational stations (larger symbols with site name: main stations used here) along with the fourth, fifth and sixth ring roads (gray lines) in Beijing, China.

[Figure]

Figure 2. (a) Hourly-mean PM2:5 concentration of CP, AOT, YZ and three-station mean in Beijing on 24 December 2015.

[Figure]

Figure 9. Doppler lidar observations of (a) vertical wind velocity, (b) horizontal wind speed, (c) wind direction, and (d) carrier–noise ratio (CNR) at IAP on 24 December 2015. The gray line indicates the time of HF passage at IAP.

[Figure]

Figure 10. Temporal variations of (a) temperature (colored contours) and wind vectors, (b) relative humidity (colored contours) and wind vectors at 15 levels on the IAP tower, and (c) vertical velocity standard deviation at 47 and 280m on the IAP tower on 24 December 2015. The gray line indicates the time of HF passage at IAP.

Based on the above analysis, we have revised and expanded the text in the discussion section, and the discussion on the indirect mechanism now reads: "*In contrast, the indirect mechanism is more intricate. It corresponds to weather scenarios with an isobaric field or weak pressure gradients. Under such mild meteorological settings, the region in front of the mountains in Jing-Jin-Ji (Beijing-Tianjin-Hebei) is prone to developing local circulations converging towards the mountain front, resulting in the accumulation of pollutants in these areas (Wang and Zhang, 2020). Here, a foehn initially appears on the leeward side, with the formation of a fast-moving,*

*low-pollution, warm-dry air mass advancing southward, encountering a slow-moving, high-pollution, cold-wet air mass from the south, potentially creating a haze front (Li et al., 2020). Given the weak nature of the foehn—characterized by low wind velocities—it fails to induce a rapid decrease or removal of pollutants. Instead, the weak pressure gradient between opposing air masses drives a "seesaw-like" exchange of air masses. The advancing warm-dry air also stabilizes the lower atmosphere by reinforcing temperature inversions above the cold air mass, in which overlying warm air increases the strength of these inversions. This enhanced atmospheric stability traps pollutants near the surface, even as the slow northward migration of the haze front continues to advect high-concentration pollutants from southern regions into relatively cleaner northern areas. Observations from Li et al. (2020) reveal that this interaction is accompanied by distinct boundary-layer evolutions: as the foehn develops, low-level northerly winds strengthen alongside subsidence motions, leading to a temporary decrease in boundary-layer aerosol concentrations; while as the foehn weakens and the haze front passes, these winds shift to weaker southerly flows, accompanied by rapid increases in boundary-layer relative humidity and pollutant concentrations—signals of enhanced moisture and pollution transport from southern sources. This indirect pathway is not a simple dilution process but involves complex interactions between frontal dynamics and boundary-layer stability. The weak foehn initially reduces surface pollutants, yet the northward retrograde of the haze front and the subsequent breakdown of vertical mixing ultimately lead to pollutant accumulation. These observations from Li et al. (2020) provide support for how mild foehn conditions, through modulating boundary-layer structure and air-mass interactions, contribute to persistent pollution patterns in the region, as evident in the "initial decrease followed by rapid surge" behavior during the second phase in Figure 11."*

Minor Comments:

Line 131-132: Pollution episodes are defined as periods with city-wide average PM2.5 >35 μg m⁻³ and a mean >75 μg m⁻³. This threshold aligns with practical air quality concerns but a brief justification or comparison with established criteria in the literature would contextualize this choice and enhance its applicability.

Thank you for the comment.

The PM2.5 thresholds employed to define pollution episodes in our study are derived from China's Ambient Air Quality Standards (GB 3095-2012) and the Technical Regulation on Ambient Air Quality Index (on trial) (HJ 633-2012). According to these regulations:

1) A 24-hour average PM2.5 concentration ≤35 μg m⁻³ corresponds to "Class I" (excellent) air quality.

2) Concentrations between 35–75 μg m⁻³ fall under "Class II" (good) air quality.

3) Values exceeding 75 μg m⁻³ indicate "Class III" or higher, signifying light pollution or more severe degradation.

The selection of 35 μg m⁻³ as the baseline threshold aligns with the transition from "excellent" to "good" air quality, while 75 μg m⁻³ reflects the onset of pollution episodes as per China's regulatory framework. This dual-threshold approach ensures consistency with national air quality management practices and provides operational relevance for policy interventions. This additional clarification is provided in the manuscript as follows: "*The PM2.5 thresholds employed*

*to define pollution episodes in our study are derived from China's Ambient Air Quality Standards (GB 3095-2012; Ministry of Environmental Protection, 2012a) and the Technical Regulation on Ambient Air Quality Index (on trial) (HJ 633-2012; Ministry of Environmental Protection, 2012b). The selection of 35 μg m⁻³ as the baseline threshold aligns with the regulatory transition from "excellent" (Class I) to "good" (Class II) air quality, while 75 μg m⁻³ reflects the onset of pollution episodes (Class III or higher) under China's air quality classification framework.*"

References:

*Ministry of Environmental Protection of the People's Republic of China. (2012). Ambient Air Quality Standards (GB 3095-2012) [Standard]. Beijing: China Environmental Science Press.*
*Ministry of Environmental Protection of the People's Republic of China. (2012). Technical Regulation on Ambient Air Quality Index (on trial) (HJ 633-2012) [Standard]. Beijing: China Environmental Science Press.*

Line 136: I think it is ERA5. Better to include the version here. The NCEP data mentioned in Line 159 is a typo? Also, The reanalysis data at 0.25° × 0.25° resolution for weather pattern classification is appropriate for large-scale patterns but may miss fine-scale topographic effects critical to foehn dynamics in Beijing's complex terrain. Have the authors considered higher-resolution datasets (e.g., regional models) to verify small-scale features? Discussing this limitation would strengthen the methodology.

Thank you for the comment. Yes, the reanalysis data in Line 136 is ERA5. We've added the version details to the revised text. The mention of "NCEP" in Line 159 was a typo. We've changed it to "ERA5" to avoid confusion.

We fully agree that the 0.25°×0.25° resolution data might miss some small-scale terrain effects, especially for processes like foehn winds in Beijing's complex topography. While we chose ERA5 for its broad applicability to large-scale patterns, we agree that higher-resolution reanalysis data (e.g., 1-km regional models) could better capture local details. We plan to explore high-resolution datasets in future work to improve the analysis.

Figure 1: White triangles are hard to see. You may need to make them thicker.
Thank you for the comment. This has been addressed by boldening the white triangles.

Line 193: How were these 22 cases identified?
Thank you for the comment. Since our initial analysis of foehn impacts on haze fronts in Beijing (Li et al., 2020), we've been closely tracking foehn events in the region. We routinely monitor temperature shifts—especially nighttime warming—at automatic weather stations (AWSs) near mountainous areas. When we spot potential foehn signals (such as abrupt air temperature increases and humidity decreases associated with mountain winds), we dig deeper by checking synoptic weather patterns, broader AWS networks, wind profile radar data, and other observations to confirm the event. Over the past five years, we've documented dozens of foehn cases. For this study, we selected 22 cases that included foehn events with strong, moderate, and weak impact levels to develop a method for identifying foehn.

Figure 2: Is CE-PAWS included in Non-NM-PAW? The national AWS is used to identify the events first. Is it possible that the regional stations have foehn events while national stations do not so you are missing some event days?

Thank you for the comment. Yes, CE-PAWS stations are categorized under Non-NM-PAWS in our analysis. We acknowledge that relying on national stations (NM-PNAWS, shown as dark blue dots in Figure 1) for initial event identification could potentially miss localized foehn events occurring only at regional stations near mountainous areas. However, the events selected in this study represent large-scale, regionally impactful foehn episodes detectable across both national and regional monitoring networks. Our prioritization of national stations stems from their 60+ years of high-quality, standardized observational data, which is critical for developing a climatologically consistent foehn identification method. While this approach may sacrifice granularity in detecting hyper-localized events, it ensures reliability and comparability for long-term trend analysis—a key focus of this work.

Figure 3: The color scale should be the same to have an easy comparison. The band-like distribution of foehn days from the northwestern mountains to the southeastern plains is visually compelling, but the paper lacks a detailed physical explanation. Is this distribution driven by the Taihang Mountains' topography, prevailing northwest winds, or a combination of factors? Linking the spatial pattern to specific atmospheric or topographic mechanisms would enhance the analysis.

Thank you for the comment.

Regarding Figure 3, we have unified the color scales across all subplots and added a color scale for topographic height (see the updated figure below).

[Figure]

Figure 3: Annual distribution of foehn days. The pink lines indicate the contour line at an elevation of 200 m.

The band-like distribution of foehn days is indeed primarily related to the orientation of the terrain. More specifically, it is associated with the alignment and topographic configuration of the Jundu Mountains, Taihang Mountains, and Xishan Mountains (Figure 1). Additionally, the frequent foehn occurrences may be linked to mountain gaps. At the CP station, the junction of the Jundu and Taihang Mountains to the north features lower terrain heights and contains

multiple gaps and valleys (see Figure 1a in Li et al., 2020, which features higher-resolution DEM data). In the Alps, many well-known foehn locations are downstream of gaps, a phenomenon explainable by hydraulic theory: asymmetric flow driven by different "reservoirs" on either side of the gap—deeper, colder subcritical flow upstream accelerates as it narrows through the gap, transitioning to thinner supercritical flow downstream, accompanied by strong subsidence and turbulent mixing. During subsidence, air warms due to adiabatic compression and experiences reduced relative humidity, leading to foehn events (Mayr et al., 2007). However, further research is needed to fully analyze the impact of gap flow on foehn events in Beijing. The corresponding text in the manuscript has been revised to: " *Figure 3 illustrates the annual cumulative distribution of foehn days at PAWSs, revealing a generally consistent horizontal distribution pattern across different years. High-frequency foehn zones are roughly aligned in a northwest-to-southeast direction. This band-like distribution of foehn days is associated with the alignment and topographic configuration of the Jundu Mountains, Taihang Mountains, and Xishan Mountains (Fig. 1). Additionally, the frequent occurrence of foehn events may be linked to topographic gaps. Specifically, the junction of the Jundu and Taihang Mountains contains multiple gaps and valleys. Studies in the Alps have shown that many foehn events occur downstream of such gaps, which is attributed to the transition of airflow from subcritical to supercritical flow as it passes through the gaps. This transition generates strong subsidence and turbulent mixing: during subsidence, air warms due to adiabatic compression and experiences reduced relative humidity, ultimately leading to foehn conditions (Mayr et al., 2007)*".

We have also added the following reference to the manuscript:

Mayr GJ, Armi L, Gohm A, et al. Gap flows: Results from the Mesoscale Alpine Programme. Quarterly Journal of the Royal Meteorological Society, 2007, 133(881): 881–896. https://doi.org/10.1002/qj.66

Line 242: Any explanations about what drives this narrowing trend?

Thank you for the comment. This inter-annual variation characteristic may be related to the climate change fluctuations of the foehn wind. Climate change can affect the frequency, intensity of the foehn wind and the scope of its impact on the ground. However, more targeted research needs to be carried out.

Table 2: While this seasonal variation is well-documented, the underlying reasons are not explored. Are winter maxima linked to stronger pressure gradients, more frequent cold fronts, or topographic amplification of downslope winds? Including a brief discussion of potential meteorological drivers would provide deeper insight into the climatology of foehn events in Beijing.

Thank you for the comment.

The wintertime prevalence of foehn events in Beijing is indeed closely associated with enhanced meteorological conditions favoring their development. As discussed in the subsequent SOM analysis, frequent influences from northerly cold air masses in winter often establish strong pressure gradients between mountainous and plain regions. This dynamic aligns with the classic mechanism for pressure gradient-driven downslope winds. Additionally, the theory of lee waves—critical for understanding foehn formation—posits that when airflow

encounters mountains with heights exceeding π/l (where l² denotes the Scorer parameter, as defined by Scorer & Klieforth, 1959), foehn conditions may emerge. The generation of lee waves relies on stable stratification in the upstream atmosphere: as air flows over mountains, it is forced upward, and under the net buoyancy effect, internal gravity waves form. When the Scorer parameter decreases rapidly with height, trapped lee waves or resonant waves occur (Scorer, 1949). In winter, periods without cold air intrusions are characterized by stable atmospheric conditions that often coincide with air pollution accumulation. Significantly, such stable stratification also facilitates the formation of lee waves and subsequent foehn events as high-pressure systems advance. This explains why our subsequent analysis shows that foehn commonly occurs during and at the end of pollution episodes, when atmospheric stability is pronounced and conducive to wave resonance over the complex topography surrounding Beijing.

We add the following content at line 260 of the original text: "*The high frequency of foehn events in winter is mainly related to the cold high-pressure systems coming from the northwest. More stable atmospheric stratifications, combined with the intrusion of cold high-pressure systems, are conducive to the formation of lee waves, which in turn generate foehn winds. These foehn events typically occur during and at the end of pollution episodes (which will be further analyzed in the subsequent sections).*"

Table 3: Can relative humidity and wind conditions be discussed in addition to temperature?
Thank you for the comment. Following your suggestion, we have included the statistical values for hourly variations in relative humidity and wind speed alongside temperature, and generated a revised Table 3 (see below). The corresponding text in the manuscript has been updated as follows: "*We analyzed their hourly temperature difference ($\triangle T$), relative humidity difference ($\triangle RH$), and wind speed difference ($\triangle WS$) on foehn days. As shown in Table 3, the median values of $\triangle T$ at these stations are highly consistent, ranging from 1.7–1.8 °C. The mean $\triangle T$ spans 2.0–2.2 °C, with the most pronounced increase observed at Station HD and the smallest at Station TZ. The maximum $\triangle T$ is greatest at TZ (11.8 °C), followed by HD (10.1 °C), and then CP (7.5 °C). When examining the 25th and 75th percentile values, half of the hourly warming instances at each station fall within a 1.3–2.6 °C range; however, the warming span for TZ is narrower than the other three stations, confined to 1.3–2.4 °C. The average values of $\triangle RH$ at each station all show a decrease, ranging from -8% to -11%. The maximum reduction of $\triangle RH$ reaches its highest value at HD (-75%) and the lowest at CP (-59%). The 25th and 75th percentiles of $\triangle RH$ fall between -14% and -3%. The average values of $\triangle WS$ at each station all show an increase, ranging from 0.4 to 0.7 m/s. The maximum value of the maximum $\triangle WS$ at each station is observed at TZ (8.8 m/s), and the minimum at HD (4.8 m/s). The 25th and 75th percentiles of $\triangle WS$ are between -0.3% and 1.4 m/s. The negative values of $\triangle WS$ may be related to the decrease in wind speed during consecutive foehn hours after the passage of the foehn.*"。

**Table 3.** Summary statistics of hourly differences in temperature ($\triangle T$, °C), relative humidity ($\triangle RH$, %), and wind speed ($\triangle WS$, m s⁻¹) at the four studied stations.

| $\triangle T$ (℃) | | | | $\triangle RH$ (%) | | | | $\triangle WS$ (m s⁻¹) | | | |
|---|---|---|---|---|---|---|---|---|---|---|---|
| CP | HD | CY | TZ | CP | HD | CY | TZ | CP | HD | CY | TZ |

| | | | | | | | | | | | | |
|---|---|---|---|---|---|---|---|---|---|---|---|---|
| max | 7.5 | 10.1 | 8.7 | 11.8 | -1 | -1 | -1 | -1 | 8.2 | 4.8 | 5.7 | 8.8 |
| min | 1.0 | 1.0 | 1.0 | 1.0 | -59 | -75 | -72 | -67 | -4.1 | -4.2 | -3.0 | -3.0 |
| median | 1.7 | 1.8 | 1.7 | 1.7 | -5 | -6 | -7 | -6 | 0.5 | 0.3 | 0.5 | 0.4 |
| mean | 2.1 | 2.2 | 2.1 | 2 | -8 | -9 | -11 | -10 | 0.7 | 0.4 | 0.6 | 0.6 |
| 25th Percentile | 1.3 | 1.3 | 1.3 | 1.3 | -10 | -12 | -14 | -13 | -0.3 | -0.2 | -0.1 | -0.2 |
| 75th Percentile | 2.6 | 2.6 | 2.5 | 2.4 | -3 | -3 | -4 | -3 | 1.4 | 1.0 | 1.2 | 1.1 |

Figure 8: It is confusing how the figure is made and more explanations in the texts would aid comprehension. It says "each episode's initiation is marked by its Local Standard Time (LST)", but the axis has larger numbers than 24. Also, the subtitle of this figure is not very clear caused by the semi-colons. The axis labels are too small to read, which also applies to some other figures (such as Figure 9).

Thank you for the comment.

The semicolons in the figure caption of Figure 8 have been replaced with colons to enhance grammatical precision.

To eliminate potential ambiguity in axis interpretation, the x-axis label has been simplified from "Hour (LST)" to "Hour", complemented by an expanded caption explanation: "Each pollution episode is aligned according to its initiation time within the 0-23 LST on the x-axis."

Corresponding explanatory text has been added in lines 329-335 of the manuscript, explicitly stating: "*Here, each pollution episode is aligned according to its initiation time within the 0-23 Local Standard Time (LST) on the x-axis, with semi-transparent filled line plots illustrating PM2.5 concentration versus time. This alignment methodology facilitates the identification of LST-dependent PM2.5 variation characteristics through composite plotting. While the terminal positions of individual episodes generally correspond to their duration (in hours), it should be noted that these plotted durations may exceed actual episode lengths in most cases, though never surpassing 24 hours.*"

Additional visual optimizations include: 1) Color modification of the plot line in Figure 8a from pink to light blue for improved chromatic discrimination; 2) Harmonization of right y-axis tick label colors with their respective axis titles in Figures 8a and 8b; 3) Systematic enlargement of critical annotations and labels across Figures 3-11 to ensure legibility.

[Figure]

Figure 8: Characteristics of pollution episodes and associated foehn occurrences. Each pollution episode is aligned according to its

initiation time within 0–23 LST on the x - axis. (a) Temporal variation of the PM 2.5 concentration during pollution episodes, with the green line representing the number of pollution episodes and the light-blue line indicating the average PM 2.5 concentration of pollution episodes. (b) Foehn occurrence during pollution episodes, with gray bars indicating the cumulative number of stations with foehn occurrences per episode (considering only the 14 plain national stations) and red scatter points representing the maximum hourly temperature change.

Line 490. The AWS-based foehn identification method is a key strength, enabling long-term analysis with widely available data. However, the paper does not discuss its potential application to other regions with similar topography (e.g., other parts of the North China Plain or mountain-adjacent cities globally). Addressing generalizability would increase the study's impact and relevance to the broader atmospheric science community.

Thank you for the comment. We have rewritten the last paragraph. Now it reads: "*The foehn identification method proposed in this study, which relies solely on surface AWS data, facilitates the identification of foehn events using long-term historical observational data. For climatological studies of foehn winds worldwide, the application of methodologies analogous to those presented herein enables the analysis of long-term observational datasets from a limited number of surface meteorological monitoring stations. This approach facilitates a deeper understanding of how foehn phenomena evolve and contribute to temperature increases in the context of global warming. Additionally, it enhances researchers' ability to investigate the relationships between foehn winds and high-impact weather phenomena, such as air pollution and heatwaves.*"

**Reviewer: 2**

The paper introduces a novel foehn identification method for the Beijing region, emphasizing seasonal and spatial patterns influenced by local topography and meteorological conditions. The potential strength of this work lies in the simplicity of the proposed method, which relies exclusively on near surface meteorological observations, making it suitable for long-term climatological applications. However, the authors do not clearly describe and discuss this methodological innovation throughout the manuscript. While the study presents a promising approach, the manuscript would benefit from improvements in clarity, methodological transparency, and deeper discussion of the results and their implications.

The study first identifies 204 pollution episodes in Beijing based on city-wide PM2.5 concentrations exceeding 75 μg/m³. A method is then applied to detect foehn events using near surface meteorological data, considering wind direction, temperature increase, humidity drops, and no precipitation. The authors cross-reference the pollution episodes with foehn occurrences and find that 137 of them overlap. These are manually classified into two types: Type I (rapid PM2.5 decrease) and Type II (slight decrease followed by a rapid increase). The classification appears to be subjective, based on visual inspection of time series. Finally, Self-Organizing Maps (SOMs) are used to identify distinct synoptic patterns associated with each foehn type.

**General comments:**

1)     The authors use PM2.5 time series during pollution episodes (defined by concentrations exceeding 75 μg/m³) to classify events into Type I and Type II. However, it is only between lines 382 and 401, in reference to Figure 11, that we get a clearer idea of what is meant by the "rapid pollutant concentration decreases" (Type I) and a "slight pollutant concentration decreases followed by a rapid increase" (Type II). Based on Figure 11, a "rapid" decrease seems to occur over approximately 6 hours, is that correct? I assume not all cases follow the same time window. Could the authors clarify how this visual classification was performed? Was any threshold defined? This point is particularly relevant if the method is to be applied to longer time series, where visual inspection alone may not be feasible.

Thank you for the comment. Based on the subjective classification results, we established the following objective classification method and ensured the consistency between the subjective and objective classification results.

A foehn event is defined as a sequence of continuous or quasi-continuous foehn hours lasting at least 2 hours, where quasi-continuity allows intervals of up to 2 hours between successive foehn hours, subsequently merged into a single event.

Foehn events were classified into two distinct types according to PM$_{2.5}$ concentration dynamics. Type I: the type with rapid decline in pollutant concentration, which is defined by the simultaneous satisfaction of two criteria:

(1)  for a foehn event, the PM$_{2.5}$ concentration change (ΔC)—defined as the concentration at the hour immediately before event initiation minus the concentration at event termination—must be negative;

(2)  a 25% reduction in PM$_{2.5}$ concentration at the event termination compared to the initial value.

Type II: the type with rapid increase in pollutant concentration or slight decrease first and then rapid increase, which is defined by the simultaneous satisfaction of two criteria:

(1) a non-negative $\Delta C$, or a negative $\Delta C$ with a terminal concentration reduction of less than 5% from the initial value;

(2) the emergence of a new $PM_{2.5}$ peak with a concentration increase of more than 25% from the initial value, occurring either before subsequent foehn events if there are any or within 24 hours after this foehn event termination if there aren't.

All identified foehn events underwent rigorous screening across pollution episodes, complemented by manual validation to ensure methodological robustness. For compatibility of self-organizing map (SOM) analysis with daily ERA5 sea-level pressure (SLP) data, only days exhibiting single-type foehn events (exclusively Type I or II) were included.

We've made expansions and revisions to the main text—you can find the detailed revised content in our response to Reviewer 1.

2)    When the authors mention that the foehn identification method was developed based on 22 representative historical cases, it is not clear whether these cases were derived from previous literature or from the same dataset used in this study. It would be helpful to clarify this point more explicitly. Were these 22 historical cases associated with pollution episodes? Were they classified as Type I, Type II or both?

Thank you for the comment. Since our initial analysis of foehn impacts on haze fronts in Beijing (Li et al., 2020), we've been closely tracking foehn events in the region. We routinely monitor temperature shifts—especially nighttime warming—at automatic weather stations (AWS) near mountainous areas. When we spot potential foehn signals (such as abrupt air temperature increases and humidity decreases associated with mountain winds), we dig deeper by checking synoptic weather patterns, broader AWS networks, wind profile radar data, and other observations to confirm the event. Over the past five years, we've documented dozens of foehn cases. For this study, we selected 22 cases encompassing foehn events with strong, moderate, and weak impact levels to develop our foehn identification method. These cases were not specifically chosen from pollution episodes; thus, not all of them are associated with pollution events. Additionally, only four cases occurred in 2020, while the remaining cases fall outside the temporal scope of the data analyzed in this paper. In future research, we will further investigate the relationship between these cases and pollution processes, as well as classify the foehn events into Type I, Type II, or both categories.

3)   I appreciate the use of standard meteorological data from AWS to identify foehn patterns, and this strength could be further emphasized throughout the text. While I value the simplicity of the approach, I wonder whether the authors have access to eddy covariance system data to quantify turbulence-related variables, such as turbulent kinetic energy (TKE) or the standard deviation of vertical velocity. These metrics could provide direct evidence of turbulence intensity and offer a more detailed view of how foehn winds enhance vertical mixing and potentially contribute to pollutant dispersion.

Thank you for the comment. We appreciate your suggestion to emphasize the use of standard AWS data for foehn identification. In the conclusion section, we have highlighted the methodology's applicability: "*The foehn identification method proposed in this study, which*

*relies solely on surface AWS data, facilitates the identification of foehn events using long-term historical observational data. For climatological studies of foehn winds worldwide, the application of methodologies analogous to those presented herein enables the analysis of long-term observational datasets from a limited number of surface meteorological monitoring stations. This approach facilitates a deeper understanding of how foehn phenomena evolve and contribute to temperature increases in the context of global warming. Additionally, it enhances researchers' ability to investigate the relationships between foehn winds and high-impact weather phenomena, such as air pollution and heatwaves."*

Regarding your question about eddy covariance data: In our previous publication (Li et al., 2020), we analyzed and discussed the standard deviation of vertical velocity (see Figure 10c, which is included in our response to Reviewer 1), and indeed observed a significant impact of foehn winds on turbulence. However, due to limitations in accessing turbulence datasets, we were unable to incorporate such analyses in the current paper. We plan to conduct in-depth analyses on this topic in future research.

**Minor comments:**

-The captions of several figures are unclear and lack essential details. I recommend that the authors carefully review each figure and revise the captions to provide complete and self-explanatory information.

Thank you for the comment. We have reviewed and optimized the captions for all figures in the manuscript. Please refer to the revised version for the updated details.

**Line 231**: the caption of Figure 3 states: "Annual distribution of foehn days", but the accompanying map appears to include a Digital Elevation Model (DEM). However, there is no legend to explain the meaning of the colors used.

Thank you for the comment. We have added a color bar for the DEM in Figure 3 and made other corresponding modifications. Please refer to our response to Reviewer 1 for the revised figure.

**Line 276:** A similar issue applies to Figure 5; the color shading is not accompanied by a legend.

Thank you for the comment. We have revised Figure 5 as well. Please refer to our response to Reviewer 1 for the updated figure.

**Line 308:** In Figure 7, what does the red square represent?

Thank you for the comment. The red square represents the average value, and we have added an explanation to the caption of Figure 7.

[Figure]

Figure 7: Temporal variations in the hourly temperature changes: (a–d) diurnal variations and (e–h) monthly variations. Box-and-whisker plots show the 25th, 50th (median), and 75th percentiles, with the red square indicating the mean value.

**Line 342**. The linear regression equation and axis labels are too small and difficult to read.
Thank you for the comment. We have made the revisions, and the revised figure is shown below.

[Figure]

Figure 9: Correlation between pollution episode duration and the foehn-to-pollution ratio.

**Line 378:** Figure 11 illustrates a case study showing the temporal evolution of PM2.5 concentrations and foehn event occurrences during a pollution episode. While the authors refer to "phases" (Phase I, Phase II, and Phase III), these phases are not marked in the figure. The dashed vertical lines might correspond to these phases, but this is not explicitly stated. The figure caption lacks clarity and does not adequately guide the reader.

Thank you for the comment. We have made the revisions, and the revised figure is shown below.

[Figure]

Figure 11. Temporal evolution of PM2.5 concentrations and foehn event occurrence during a pollution episode. Primary Y-axis (left): PM2.5 concentration (black line) and hourly PM2.5 concentration change (orange line).

Secondary Y-axis (right): Number of stations experiencing foehn events (blue bars). Three distinct phases of the pollution episode are highlighted with gray-shaded areas, corresponding to different foehn-type classifications.

**Line 398**: The term "weak pressure gradients" is used, but no numerical values are provided (e.g., 1–3 hPa over 500 km?).

Thank you for the comment. We have revised the relevant paragraph, and the revised text is as follows: "*For Type I (depicted in Fig. 12), a consistent high-pressure system is observed northwest of Beijing, accompanied by a pressure gradient directed from northwest to southeast. To quantify the pressure gradient, we use the pressure difference (△P) between the center of the Beijing Plain (the ring road's center in Fig. 1) and a point 300 km northwest of this center. Notably, SOM types SOM2 and SOM4, which feature strong high-pressure systems and pronounced pressure gradients (△P > 6 hPa), jointly account for 36.25% of occurrences. These conditions are frequently associated with the passage of cold fronts, facilitating the rapid dispersion of pollutants. SOM3 and SOM5 share similar pressure patterns to SOM2 and SOM4 but exhibit weaker pressure gradients (3 hPa < △P < 6 hPa), collectively representing 30% of instances. The weakest pressure gradients (△P ≈ 3 hPa) are observed in SOM1 and SOM6 types, together comprising 33.75% of cases. For Type II, Beijing is predominantly located within a near-isobaric field in SOM1 and SOM4, while SOM2 and SOM3 show weak pressure gradients (△P ≈ 3 hPa) to the northwest or west of the city.*"

**Line 403:** Throughout the manuscript, the authors frequently omit units, symbols. For example, in Figure 12, the unit of sea level pressure is not indicated.

Thank you for the comment. We have checked the entire manuscript and added missing units and symbols. In Figure 12, we included the unit for sea level pressure.

**Line 411 – 416**: The authors acknowledge the challenge of distinguishing foehn-induced warming from other non-foehn processes, such as solar radiation or warm air advection. While they suggest that this issue could be mitigated by incorporating detailed analyses of wind fluctuations and upstream–downstream consistency (in Zhang and Li, 2024), such an approach is not actually applied in the current study. As a result, the risk of misclassifying non-foehn warming as foehn, especially at downstream locations, remains unresolved. Explicitly addressing this limitation, or testing the proposed checks within the study itself, would greatly improve the credibility and robustness of the results.

Thank you for the comment. Considering your comment and that of Reviewer 1, we have rewritten the first paragraph in the discussion section as follows: "*The identification of foehn winds using AWS data requires careful differentiation from other warming mechanisms through characteristic meteorological signatures. Foehn events are distinguished by abrupt temperature increases (>1℃/hour), simultaneous humidity drops, and sustained winds aligned with mountain-plain airflow patterns (typically W/NW in Beijing), contrasting sharply with warm front-associated warming that exhibits gradual temperature rise, moisture increases, and E/SE winds. While tropical cyclone peripheral warming and anticyclonic subsidence demonstrate even lower thermal gradients and broader spatial impacts, our methodology employs wind-direction verification and thermal thresholds to effectively exclude these phenomena. Foehn warming differs significantly from solar radiation-induced warming in meteorological element change characteristics. Foehn warming is characterized by rapid short-term temperature surges accompanied by abrupt wind speed increases, sharp humidity drops, and*

*clear directional movement from mountains to plains. In contrast, solar radiation warming lacks instantaneous abrupt changes in meteorological elements, with wind directions primarily influenced by local wind systems. In Beijing's plain areas without large-scale weather systems or foehn effects, mountain-valley and mountain-plain wind systems dominate, causing significant diurnal variations in near-surface wind directions: nocturnal winds blow from mountains to plains, while daytime winds reverse to blow from plains to mountains. At night without solar radiation, our foehn identification method effectively detects foehn at mountain-proximal stations due to the unidirectional mountain-to-plain airflow consistent with foehn movement. During daytime solar radiation warming without foehn, valley winds and plain-to-mountain winds cause plain station wind directions to point toward mountains (opposite to foehn direction), thereby preventing false foehn detection. However, downstream foehn propagation may lead to misidentification issues. Warming mechanisms at downstream stations involve a combination of foehn-related processes (advection and lee wave subsidence) and solar radiation heating (daytime only). Foehn advection may transport locally warmer air (e.g., urban heat island) downstream, resulting in overestimation of foehn occurrence by our method. Daytime solar radiation exaggerates foehn warming magnitude, while foehn-induced cloud-free or few-cloud conditions ("foehn clearance", Hoinka, 1985a) further enhance solar radiation, creating a coupled direct-indirect foehn effect. Although strictly meteorological criteria might classify such events as overestimations, the observed thermal enhancements remain fundamentally tied to foehn-initiated processes, warranting their inclusion in broader impact assessments of foehn phenomena.*"